# The rarefied (non-continuum) conditions of tracer particle transport in soils, with implications for assessing the intensity and depth dependence of mixing from geochronology

David Jon Furbish[1,2], Rina Schumer[3], and Amanda Keen-Zebert[4]

[1]Department of Earth and Environmental Sciences, Vanderbilt University, Nashville, Tennessee, USA
[2]Department of Civil and Environmental Engineering, Vanderbilt University, Nashville, Tennessee, USA
[3]Division of Hydrologic Sciences, Desert Research Institute, Reno, Nevada, USA
[4]Division of Earth and Ecosystem Sciences, Desert Research Institute, Reno, Nevada, USA

**Correspondence:** David Furbish (david.j.furbish@vanderbilt.edu)

**Abstract.** We formulate tracer particle transport and mixing in soils due to disturbance driven particle motions in terms of the Fokker-Planck equation. The probabilistic basis of the formulation is suitable for rarefied particle conditions, and for parsing the mixing behavior of extensive and intensive properties belonging to the particles rather than to the bulk soil. The significance of the formulation is illustrated with the examples of vertical profiles of expected Beryllium-10 ($^{10}$Be) concentrations and

optically stimulated luminescence (OSL) particle ages for the benchmark situation involving a one-dimensional mean upward soil motion with nominally steady surface erosion in the presence of either uniform or depth dependent particle mixing, and varying mixing intensity. The analysis, together with Eulerian-Lagrangian numerical simulations of tracer particle motions, highlight the significance of calculating ensemble expected values of extensive and intensive particle properties, including higher moments of particle OSL ages, rather than assuming *de facto* a continuum-like mixing behavior. The analysis and

results offer guidance for field sampling and for describing the mixing behavior of other particle and soil properties. Profiles of expected $^{10}$Be concentrations and OSL ages systematically vary with mixing intensity as measured by a Péclet number involving the speed at which particles enter the soil, the soil thickness, and the particle diffusivity. Profiles associated with uniform mixing versus a linear decrease in mixing with depth are distinct for moderate mixing, but become similar with either weak mixing or strong mixing; uniform profiles do not necessarily imply uniform mixing.

## 1  Introduction

Soils on Earth's surface are granular materials consisting of polymineralic clasts and individual mineral grains, organic matter and live biota. These materials experience patchy, intermittent mixing motions associated with disturbances due to bioturbation (Darwin, 1881; Shaler, 1891; Gabet, 2000; Reichman and Seabloom, 2002; Meysman et al., 2006; Wilkinson et al., 2009; Covey et al., 2010; Astete et al., 2015), the effects of frost and ice growth and thaw (Branson, 1992; Matsuoka and Moriwaki, 1992;

Auzet and Ambroise, 1996; Branson et al., 1996; Harris et al., 1997; Matsuoka, 1998; Anderson, 2002), and the swelling and shrinking of certain minerals with wetting and drying (Eyles and Ho, 1970; Fleming and Johnson, 1975). In addition, these soil materials may undergo mixing motions in relation to the chronic creation and relaxation of disordered granular structures

(Hsiau and Hunt, 1993; Utter and Behringer, 2004; Fan et al., 2015) associated with granular creep (Houssais et al., 2017; Ferdowsi et al., 2018).

Soil particle mixing is a key process in soil formation (Shaler, 1891; Birkeland, 1984; Wilkinson et al., 2009) and in its associated ecological role of "modifying geochemical gradients, redistributing food resources, viruses, bacteria, ...and eggs" (Meysman et al., 2006), as well as being responsible for redistributing substances, including contaminants, attached to particles (Cousins et al., 1999; Covey et al., 2010; Astete et al., 2015). Moreover, the idea of disturbance driven transport and mixing of soil particles is central to current treatments of soil creep (Culling, 1963; Roering et al., 1999, 2002; Gabet, 2000; Anderson, 2002; Gabet et al., 2003; Furbish, 2003; Roering, 2004; Furbish et al., 2009, 2018a), the slow but steady bulk motion of soils on hillslopes, where the influence of gravity gives a downslope bias to particle motions. Because of the significance of soil particle mixing in numerous problems spanning ecological to geomorphic timescales, there is a continuing, compelling need to fully clarify the kinematics, and eventually the mechanical basis, of soil particle motions during transport and mixing (Furbish et al., 2009b, 2018a, 2018b; BenDror and Goren, 2018; Ferdowsi et al., 2018).

Currently it is not possible to directly measure disturbance driven particle motions and associated mixing in the setting of a natural soil (although this is entirely possible in experiments and numerical simulations of granular creep (Utter and Behringer, 2004; Kamrin and Koval, 2012; Fan et al., 2015)). Moreover, we do not yet have a mechanical theory to describe these motions given the complexity — notably the biotic complexity — of phenomena involved in disturbances and associated particle displacements (Furbish et al., 2009b, 2018a). Thus, as in studies of particle mixing associated with marine bioturbation (Boudreau, 1986a, 1986b; Boudreau and Imboden, 1987; Teal et al., 2008; Lecroart et al., 2010), a key strategy to clarify the nature of particle motions and mixing in soils involves using tracer particles identified by specific physical or chemical properties. Two tracer properties have emerged in the field of geomorphology as being of particular interest: *in situ* cosmogenic radionuclide (CRN) concentrations and optically stimulated luminescence (OSL) particle ages (Granger and Riebe, 2014; Heimsath et al., 2002; Johnson et al., 2014). Cosmogenic nuclides continually accumulate within minerals due to cosmic ray interactions with mineral atom nuclei, for example, producing $^{10}$Be from spallation of oxygen nuclei. Using luminescence systematics, the time elapsed since luminescence-sensitive particles were last exposed to light or heat at the soil surface is estimated from the luminescence signal that accumulates within the crystal lattice in response to a combination of ionizing radiation emitted from the decay of radioactive elements in the surrounding soil and cosmic radiation (Rhodes, 2011). Particles that accumulate CRN atoms or luminescence signals during their complex motions within soils — thereby serving as tracer particles — are naturally occurring (as opposed to being "seeded") and therefore behave mechanically the same as other soil particles. As a consequence, CRN and OSL tracer particles are particularly relevant in assessing particle mixing over timescales of soil formation and transport in the context of landform and landscape evolution.

Building from the pioneering work of Lal (1991) concerning the relation between rock erosion rates and the *in situ* production of cosmogenic radionuclides, vertical profiles of CRN concentrations in soils and underlying saprolite are now used to calculate soil production rates (e.g. Heimsath et al., 1997, 2000, 2005, 2012; Small et al., 1999; Anderson, 2002; Wilkinson et al., 2005) as well as to infer the intensity of soil particle mixing in the presence of mechanical and chemical erosion (Small et al., 1999; Schaller et al., 2009; Granger and Riebe, 2014; Furbish et al., 2018b). Similarly, profiles of particle OSL ages are used to assess

particle mixing (Heimsath et al., 2002; Wilkinson and Humphreys, 2005; Johnson et al., 2014; Furbish et al., 2018b). Because profiles of CRN concentrations and OSL ages inform descriptions of soil transport and interpretations of the delivery of CRNs to channels (Heimsath et al., 2002; Anderson, 2015; Furbish et al., 2018b), and associated interpretations of erosion rates at catchment scales (e.g., Brown et al., 1995; Bierman and Steig, 1996; Granger et al., 1996; Granger and Riebe, 2014; Granger and Schaller, 2014; Lukens et al., 2016), there is merit in further clarifying what these profiles reveal about particle mixing in soils.

It is now conventional to conceptualize certain soil particle mixing motions as a diffusion-like process (Furbish et al., 2009b, 2018a, 2018b; Campforts et al., 2016), building from the pioneering work of Culling (1963), who first pointed to the idea that soil particles undergo Gaussian diffusion in response to small disturbances. Various studies have thus appealed to some form of a diffusion equation or an advection-diffusion equation (Cousins et al., 1999; Covey et al., 2010; Stang et al., 2012; Johnson et al., 2014; Furbish et al., 2009b, 2018a, 2018b; Astete et al., 2015; Campforts et al., 2016; Gray, 2018) to describe transport and mixing for comparison with measured vertical profiles of tracer particles in soils, notably including *in situ* CRN concentrations and particle OSL ages. But herein arises a need for caution, and clarity.

As described in Section 2, natural tracer particles — quartz particles in particular — occur under rarefied conditions, where it is unclear that a description of particle mixing based on a diffusion or advection-diffusion equation formulated for continuum conditions is satisfactory. Moreover, we often are interested in the transport of quantities that are associated with the particles, and are not in themselves subject to advection and diffusion as normally envisioned to occur in a continuum. This includes particle CRN concentrations and OSL ages. Rather, such quantities might experience advection and diffusion, but only indirectly via the motions of the particles with which the quantities are associated. Within this context, our objectives in this mostly theoretical contribution are five.

First, we illustrate why quartz tracer particles in soils experience transport and mixing under rarefied (non-continuum) conditions, and why it therefore becomes important to treat transport and mixing probabilistically, in a manner that formally appeals to the statistical mechanics idea of ensemble expected (average) quantities. Our focus on quartz particles is purposeful, as these are ideal targets for *in situ* production of $^{10}$Be atoms, and for accumulating OSL signals. Second, we illustrate how the probabilistic basis of the Fokker-Planck equation, versus an "ordinary" continuum-like advection-diffusion equation, is well suited to the problem of rarefied conditions. Third, because extensive and intensive properties such as particle volume, $^{10}$Be concentration and OSL age "belong" to individual particles, not to the bulk soil, we illustrate why the probabilistic basis of the Fokker-Planck equation is suitable for parsing the mixing behavior of these properties — as opposed to assuming *de facto* a continuum-like mixing behavior in which these properties are assigned to the bulk soil. Fourth, we provide complementary numerical analyses that reveal important information not readily apparent in the analytical formulations, including an illustration of the variability in $^{10}$Be concentrations and OSL ages of individual particles in soils, with implications for interpreting field-based measurements. This part of the paper highlights a benchmark situation involving a one-dimensional mean upward soil motion with nominally steady surface erosion in the presence of either uniform or depth dependent particle mixing, and varying mixing intensity. Fifth, we use the results for this benchmark case in relation to published field-based measurements to suggest constraints on assessing the intensity and depth dependence of mixing.

Note that in the formulations presented below, we use full functional notation throughout. This provides clarity in how random variables, parameters, and moments of random variables depend on position and time, as well as how random variables might covary.

## 2    Rarefied versus continuum particle conditions in soils

Quartz particles targeted in sampling for $^{10}$Be analysis typically are within the range of $0.25 - 0.50$ mm; but sometimes grains as small as $0.125$ mm and as large as $0.85$ mm or $1$ mm are sampled from quartz-poor source materials (Gosse and Phillips, 2001; Morgan et al., 2011; Shakun et al., 2018). Quartz particles targeted for single-grain OSL analysis typically are within the range of $0.35 - 0.425$ mm (e.g., Heimsath et al., 2002; Johnson et al., 2014), but smaller grains sometimes are used. Thus, neglecting aeolian inputs, target grains represent a subset of the total population of quartz grain sizes in soils released from parent bedrock during soil formation. In the following discussion we consider for illustration a single particle size, with recognition that the ideas extend to other particles.

Consider a soil element with dimensions $XYh$, where $X = Y = h = 1$ m, residing on a soil-mantled hillslope (Figure 1). If in the ideal this element contains uniform particles of diameter $d = 1$ mm that are approximately closely packed, then the

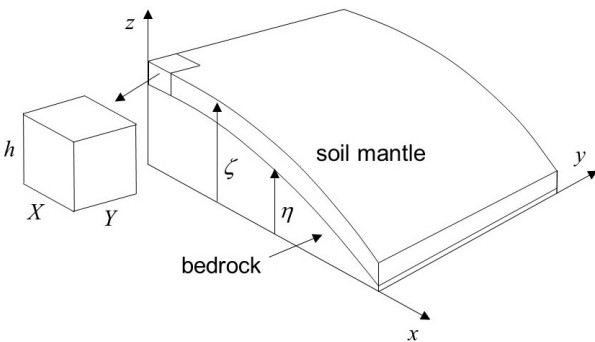

**Figure 1.** Definition diagram of soil-mantled hillslope with mechanically active soil thickness $h = \zeta - \eta$, and cutout soil element with dimensions $XYh$. Bedrock material is continually transformed into soil by chemical and mechanical processes, and soil particles are transported downslope by creep or surface erosion.

total number of particles in the soil element is O($10^9$). Each cubic centimeter contains O($10^3$) particles. The average spacing is approximately equal to one particle diameter, and the geometrical mean free path $\lambda$ (Furbish et al., 2009b) is a fraction of the particle diameter. For comparison, the number of molecules in a cubic centimeter of air, a continuum material at ordinary pressure-temperature conditions, is O($10^{19}$). The mean free path, which varies inversely with the molecular collision frequency or number density, is O($10^{-7}$) m, approximately $10^3$ larger than the effective molecular diameter. Assuming the continuum hypothesis is satisfied for a value of the Knudsen number $Kn = \lambda/L \leq 0.01$, then the averaging length scale $L$ defining a continuum physical point for air is O($10^{-5}$) m, far smaller than most scales of interest in treating particle transport and mixing in air.

For a soil developed from granitic bedrock, 20% to 60% of the volume of particles are quartz particles, some larger than 1 mm in diameter and many smaller. Per unit volume, the number of quartz particles targeted in sampling for [10]Be analysis thus is generally smaller than the close-packed value of $O(10^3)$ cm$^{-3}$ estimated above, and the average spacing may be on the order of millimeters to a centimeter or more. For example, in practical terms, [10]Be analysis requires about 10 g of quartz. Assuming 0.5 mm grains, this represents ~60,000 grains. For soils formed on granitic bedrock, one typically samples at least one liter of soil for 10 g of quartz. This translates to $n \sim 10 - 100$ grains per cubic cm. The associated geometrical mean free path is about $1 - 10$ cm, and the average spacing $\lambda_s \sim (cm^3/n)^{1/3}$ is 0.2 – 0.5 cm. Although the behavior of tracer particles is unlike gas particle kinetics, we nonetheless can use these quantities in analogy with the mean free path. Conservatively using the average spacing as a suitable measure of the particle number density, then to satisfy the Knudsen condition of $Kn \leq 0.01$ in order to appeal to a continuum description of particle behavior, the averaging length scale $L$ may approach the soil thickness. This condition is exacerbated if the parent bedrock is quartz poor. In addition, a small fraction — only a few percent — of quartz particles initially released from bedrock are sensitive to OSL and develop a less-than-saturated luminescence signal following exposure to sunlight or heat at the soil surface. Thus, tracer particles identified as those possessing a finite OSL age (Heimsath et al., 2002; Johnson et al., 2014) may be highly rarefied.

We therefore must admit at the outset that the number concentration of target quartz particles does not necessarily satisfy the continuum hypothesis. Nonetheless, we wish to use continuum-like formulations of transport and mixing of particle concentrations and associated quantities, that is, where particle concentrations, [10]Be concentrations and particle OSL ages may be viewed as continuously differentiable functions of position and time. In order to justifiably do this, we therefore appeal to the idea of an ensemble of particle configurations, a statistical mechanics idea designed to treat rarefied particle conditions.

For an element of soil with dimensions $XYh$ (Figure 1), let $f_z(z,t)$ denote the probability density function of particle positions $z$ within the element. Thus, $f_z(z,t)dz$ represents the probability that a particle is located within the small interval $z$ to $z + dz$ at time $t$. This represents an ensemble expected value, as follows. We envision, as did Gibbs (1902), a great number (an ensemble) of nominally identical but independent systems, each containing a large number $N$ of particles and behaving in a statistically similar manner with respect to transport and mixing. The expected number of particles within the interval $z$ to $z + dz$ in any system (realization) at time $t$ may vary from one system to another. However, we then imagine taking the expected value over the ensemble (Kittel, 1958), akin to ensemble Reynolds averaging (Monin and Yaglom, 1971). This represents the expected number of particles within $dz$, namely, $Nf_z(z,t)dz$, where $f_z(z,t)$ now is interpreted as the ensemble expected density. Moreover, we may assume that $f_z(z,t)$ is a smooth, continuous function. Further details regarding rarefied versus continuum conditions and ensemble averaging are provided in Appendix A.

In the developments below, we also consider joint probability density functions, for example, the joint density $f_{V_p,n_p,z}(V_p,n_p,z,t)$ of individual particle volumes $V_p$, [10]Be atom number concentrations $n_p$ and positions $z$. We similarly assume that these represent ensemble expected densities with respect to $z$. In principle, therefore, we are considering the expected concentration of particles and associated properties within any small interval $z$ to $z + dz$ in a soil element with dimensions $XYh$ (Figure 1), where averaging is over an ensemble of nominally identical but independent systems. In practical terms, one hopes to sample over an area $XY$ such that the number of particles within any small interval $z$ to $z + dz$ is sufficiently large to provide reason-

able estimates of ensemble averaged values, where these estimates vary approximately smoothly over $z$ and average over the effects of patchy, intermittent particle motions. However, this cannot be known *a priori*, a point to which we return below.

## 3 Formulation

### 3.1 Tracer Particles

Consider a set of tracer particles that are undergoing transport and mixing within a soil. Here we initially restrict this set to chemically resistant quartz particles. Nonetheless, this set could consist of particles defined by other mineralogies; or it could be defined as the subset of quartz particles of a given size that possess a specified [10]Be concentration or finite OSL age. For simplicity, and in anticipation of further analyses below, we focus on one-dimensional motions parallel to the $z$ axis.

As above, let $f_z(z, t)$ denote the probability density function of tracer particle positions $z$. Following Furbish et al. (2009b,

2018a, 2018b), this density satisfies a Fokker-Planck equation of the form

$$\frac{\partial f_z(z,t)}{\partial t}$$

$$= -\frac{\partial}{\partial z}\left[w_p(z,t)f_z(z,t) - \kappa_z(z,t)\frac{\partial f_z(z,t)}{\partial z}\right],\tag{1}$$

where $w_p(z,t)$ denotes the ensemble averaged particle velocity (sometimes referred to as the "drift speed") and $\kappa_z(z,t)$ denotes

the ensemble averaged particle diffusivity. Specifically, let $r = z(t + dt) - z(t)$ denote a particle displacement during the small interval of time $dt$. Then let $f_r(r; z, t)$ denote the probability density function of displacements $r$. The particle velocity $w_p(z,t)$ is then defined kinematically as

$$w_p(z,t) = \lim_{dt \to 0} \frac{a(z,t)}{dt} \int_{-\infty}^{\infty} r f_r(r;z,t)\,dr\,,\tag{2}$$

and the particle diffusivity $\kappa_z(z,t)$ is defined as

$$\kappa_z(z,t) = \lim_{dt \to 0} \frac{a(z,t)}{2dt} \int_{-\infty}^{\infty} r^2 f_r(r;z,t)\,dr\,,\tag{3}$$

where $a(z,t)$ denotes the particle activity probability, effectively the proportion of time that particles are in motion (Furbish et al., 2009a, 2009b, 2016, 2018a).

This formulation assumes Gaussian diffusion of particles. Interestingly, Culling (1963) first pointed to the idea that soil particles undergo Gaussian diffusion in association with particle concentration gradients, in response to small disturbances.

Culling developed his ideas from kinetic theory and statistical mechanics, borrowing the description of Brownian motion due to Einstein (1905) and the formulation of a particle diffusion-like equation due to Chandrasekhar (1943), both of which start from the Master equation (Risken, 1984; Ebeling and Sokolov, 2005; Furbish et al 2009a, 2009b). Culling's formulation has

for decades provided the inspiration for conceptualizing what now are referred to as "disturbance driven" particle motions associated with bioturbation, freeze-thaw cycles, etc. (Darwin, 1881; Shaler, 1891; Eyles and Ho, 1970; Fleming and Johnson, 1975; Matsuoka and Moriwaki, 1992; Auzet and Ambroise, 1996; Harris et al., 1997; Matsuoka, 1998; Gabet, 2000; Anderson, 2002; Reichman and Seabloom, 2002; Meysman et al., 2006; Wilkinson et al., 2009; Covey et al., 2010; Astete et al., 2015), and numerous authors have applied some form of a diffusion equation to describe transport and mixing of soil particles (Cousins et al., 1999; Furbish et al., 2009b, 2018a, 2018b; Covey et al., 2010; Johnson et al., 2014; Astete et al., 2015; Campforts et al., 2016; Gray, 2018).

We emphasize that Eq. (1) is basically an advection-diffusion equation. As written, it is purely kinematic, as nothing is specified mechanically about the velocity $w_p(z,t)$ or the diffusivity $\kappa_z(z,t)$. In this view, the ideas of particle advection and diffusion are purely probabilistic constructs based on the first and second moments of the particle displacements $r$ (Furbish et al., 2016, 2018a), as in Eq. (2) and Eq. (3). As a description of the time evolution of the probability density $f_z(z,t)$ of particle positions $z$, advection and diffusion in Eq. (1) refer to fluxes of probability. This means that, for a great number of particles within the soil element $XYh$, each particle "carries" a small, finite amount of probability with it as it moves over $z$. Moreover, despite the fact that Eq. (1) has the continuous form of a continuum advection-diffusion equation, Eq. (1) does not necessarily imply a continuum behavior. Only if conditions satisfy the continuum hypothesis can Eq. (1) be reinterpreted as an ordinary advection-diffusion equation describing transport and mixing in an individual (continuum) realization. For rarefied conditions, however, Eq. (1) represents the ensemble expected behavior, not necessarily what happens in an individual realization (Appendix A). We elaborate this point below in relation to expected particle positions $z$, [10]Be concentrations and OSL ages.

We reemphasize a point made above, that currently it is not possible to directly measure particle displacements $r$ and the associated probability density $f_r(r; z, t)$ in the setting of a natural soil. Nor is it possible to directly calculate the activity probability $a(z, t)$. Thus, in the absence of a mechanical theory to describe these displacements, indirect measures of particle mixing behavior as reflected by profiles of [10]Be concentrations and particle OSL ages are particularly valuable. Namely, any kinematic formulation of particle motions and mixing, specifically the underlying assumptions of the formulation, must be judged by its consistency with these profiles. In this vein, assuming Gaussian mixing is parsimonious, as an initial step, and in the absence of evidence of non-Gaussian behavior (Furbish et al., 2018a). This is essentially the same strategy adopted in early statistical mechanics, that the veracity of the fundamental assumption of equally probable microstates (Gibbs, 1902) only can be "tested" against experimental outcomes (Tolman, 1938). Moreover, we suggest that a Gaussian formulation of mixing possesses the right granularity to accommodate uncertainty that goes with field sampling of soils. That is, this formulation captures the essence of particle mixing behavior that can be tested within the current capabilities of field-based sampling and measurements of [10]Be concentrations and OSL ages.

## 3.2 Expected $^{10}$Be concentrations

### 3.2.1 Conservation of $^{10}$Be atoms

Let $f_{V_p,n_p,z}(V_p,n_p,z,t)$ denote the joint probability density function of particle volumes $V_p$, $^{10}$Be atom number concentrations $n_p$ and positions $z$. For particles with a given volume $V_p$ and concentration $n_p$, and momentarily neglecting the production and decay of $^{10}$Be atoms, the density $f_{V_p,n_p,z}(V_p,n_p,z,t)$ satisfies a Fokker-Planck equation of the form

$$\frac{\partial f_{V_p,n_p,z}(V_p,n_p,z,t)}{\partial t}$$

$$= -\frac{\partial}{\partial z}\left[w_p(z,t)f_{V_p,n_p,x}(V_p,n_p,x,t)\right.$$

$$\left. -\kappa_z(z,t)\frac{\partial f_{V_p,n_p,z}(V_p,n_p,z,t)}{\partial z}\right]. \tag{4}$$

We now define a conditional joint probability density function of volumes $V_p$ and concentrations $n_p$, namely,

$$f_{V_p,n_p|z}(V_p,n_p|z,t) = \frac{f_{V_p,n_p,z}(V_p,n_p,z,t)}{f_z(z,t)}. \tag{5}$$

Multiplying both sides of Eq. (5) by the product $V_p n_p$, rearranging, and integrating with respect to $V_p$ and $n_p$,

$$f_z(z,t)\int_0^\infty\int_0^\infty V_p n_p f_{V_p,n_p|z}(V_p,n_p|z,t)\,\mathrm{d}n_p\,\mathrm{d}V_p$$

$$= \int_0^\infty\int_0^\infty V_p n_p f_{V_p,n_p,z,t}(V_p,n_p,z,t)\,\mathrm{d}n_p\,\mathrm{d}V_p. \tag{6}$$

Note that the product $V_p n_p$ is equal to the number of $^{10}$Be atoms within a particle of volume $V_p$.

The double integral on the left side of Eq. (6) defines the expected value of the product $V_p n_p$, that is, $\overline{V_p n_p}(z,t)$. Thus,

$$f_z(z,t)\overline{V_p n_p}(z,t)$$

$$= \int_0^\infty\int_0^\infty V_p n_p f_{V_p,n_p,z,t}(V_p,n_p,z,t)\,\mathrm{d}n_p\,\mathrm{d}V_p. \tag{7}$$

If, however, $V_p$ and $n_p$ are independent, then $\overline{V_p n_p}(z,t) = \overline{V}_p(z,t)\overline{n}_p(z,t)$. More formally, if the particles are small and within a limited size range, we may assume that $V_p$ and $n_p$ are independent. In this case, $f_{V_p,n_p|z}(V_p,n_p|z,t) = f_{V_p|z}(V_p|z,t)f_{n_p|z}(n_p|z,t)$, and we rewrite Eq. (6) as

$$f_z(z,t)\int_0^\infty V_p f_{V_p|z}(V_p|z,t)\,\mathrm{d}V_p\int_0^\infty n_p f_{n_p|z}(n_p|z,t)\,\mathrm{d}n_p$$

$$= \int\limits_0^\infty \int\limits_0^\infty V_p n_p f_{V_p,n_p,z,t}(V_p,n_p,z,t)\, \mathrm{d}n_p \, \mathrm{d}V_p\,. \tag{8}$$

Evaluating the integrals on the left side of Eq. (8) then yields

$$f_z(z,t)\overline{V}_p(z,t)\overline{n}_p(z,t)$$

$$= \int\limits_0^\infty \int\limits_0^\infty V_p n_p f_{V_p,n_p,z,t}(V_p,n_p,z,t)\, \mathrm{d}n_p \, \mathrm{d}V_p\,. \tag{9}$$

where $\overline{V}_p(z,t)$ is the expected (average) particle volume and $\overline{n}_p(z,t)$ is the expected particle $^{10}$Be concentration. We use these results momentarily.

We now multiply Eq. (4) by the product $V_p n_p$ and integrate with respect to $V_p$ and $n_p$, namely,

$$\int\limits_0^\infty \int\limits_0^\infty V_p n_p \frac{\partial f_{V_p,n_p,z}(V_p,n_p,z,t)}{\partial t}\, \mathrm{d}n_p \, \mathrm{d}V_p$$

$$= -\int\limits_0^\infty \int\limits_0^\infty V_p n_p \frac{\partial}{\partial z}\Bigg[ w_p(z,t) f_{V_p,n_p,z}(V_p,n_p,z,t)$$

$$-\kappa_z(z,t)\frac{\partial f_{V_p,n_p,z}(V_p,n_p,z,t)}{\partial z}\Bigg]\, \mathrm{d}n_p \, \mathrm{d}V_p\,. \tag{10}$$

15  Noting that the random variables $V_p$ and $n_p$ are not functions of time $t$ or position $z$, and using Leibniz's rule, Eq. (10) may be written as

$$\frac{\partial}{\partial t}\Bigg[ \int\limits_0^\infty \int\limits_0^\infty V_p n_p f_{V_p,n_p,z}(V_p,n_p,z,t)\, \mathrm{d}n_p \, \mathrm{d}V_p\Bigg]$$

$$= -\frac{\partial}{\partial z}\Bigg[ w_p(z,t)\int\limits_0^\infty \int\limits_0^\infty V_p n_p f_{V_p,n_p,z}(V_p,n_p,z,t)\, \mathrm{d}n_p \, \mathrm{d}V_p\Bigg]$$

$$+\frac{\partial}{\partial z}\Bigg( \kappa_z(z,t)$$

$$\cdot \frac{\partial}{\partial z}\Bigg[ \int\limits_0^\infty \int\limits_0^\infty V_p n_p f_{V_p,n_p,z,t}(V_p,n_p,z,t)\, \mathrm{d}n_p \, \mathrm{d}V_p\Bigg]\Bigg)\,. \tag{11}$$

Using Eq. (7), this becomes

$$\frac{\partial}{\partial t}\left[f_z(z,t)\overline{V_p n_p}(z,t)\right] = -\frac{\partial}{\partial z}\left[w_p(z,t)f_z(z,t)\overline{V_p n_p}(z,t)\right]$$

$$+\frac{\partial}{\partial z}\left(\kappa_z(z,t)\frac{\partial}{\partial z}\left[f_z(z,t)\overline{V_p n_p}(z,t)\right]\right), \tag{12}$$

and using Eq. (9),

$$\frac{\partial}{\partial t}\left[f_z(z,t)\overline{V}_p(z,t)\overline{n}_p(z,t)\right]$$

$$= -\frac{\partial}{\partial z}\left[w_p(z,t)f_z(z,t)\overline{V}_p(z,t)\overline{n}_p(z,t)\right]$$

$$+\frac{\partial}{\partial z}\left(\kappa_z(z,t)\frac{\partial}{\partial z}\left[f_z(z,t)\overline{V}_p(z,t)\overline{n}_p(z,t)\right]\right). \tag{13}$$

We now turn to the production and decay terms to be added to Eq. (12) or Eq. (13).

### 3.2.2 Production and decay of $^{10}$Be atoms

In the absence of advection and diffusion, the joint probability density $f_{V_p,n_p,z}(V_p,n_p,z,t)$ satisfies a statement of conservation of probability having the form of an advection equation with respect to the $n_p$ domain, namely,

$$\frac{\partial f_{V_p,n_p,z}(V_p,n_p,z,t)}{\partial t}$$

$$= -P(z,t)\frac{\partial f_{V_p,n_p,z}(V_p,n_p,z,t)}{\partial n_p}, \tag{14}$$

where the advective speed $P(z,t) = \mathrm{d}n_p/\mathrm{d}t$ is the rate of production of $^{10}$Be atoms per unit particle volume. Multiplying Eq. (14) by the product $V_p n_p$ and using the product rule leads to

$$V_p n_p \frac{\partial f_{V_p,n_p,z}(V_p,n_p,z,t)}{\partial t}$$

$$= -P(z,t)V_p n_p \frac{\partial f_{V_p,n_p,z}(V_p,n_p,z,t)}{\partial n_p}$$

$$= -P(z,t)\frac{\partial}{\partial n_p}\left[V_p n_p f_{V_p,n_p,z}(V_p,n_p,z,t)\right]$$

$$+P(z,t)V_p f_{V_p,n_p,z}(V_p,n_p,z,t). \tag{15}$$

Because the product $V_p n_p f_{V_p,n_p,z}(V_p, n_p, z, t)$ represents a proportion of all $^{10}$Be atoms in the soil column, we may at this point add the effect of radioactive decay, so that Eq. (15) becomes

$$V_p n_p \frac{\partial f_{V_p,n_p,z}(V_p, n_p, z, t)}{\partial t}$$

$$= -P(z,t) \frac{\partial}{\partial n_p} \left[ V_p n_p f_{V_p,n_p,z}(V_p, n_p, z, t) \right]$$

$$+ P(z,t) V_p f_{V_p,n_p,z}(V_p, n_p, z, t)$$

$$- \lambda V_p n_p f_{V_p,n_p,z}(V_p, n_p, z, t), \tag{16}$$

where $\lambda$ denotes the decay constant.

In turn, integrating Eq. (16) with respect to $V_p$ and $n_p$, and using Eq. (9),

$$\frac{\partial}{\partial t} \left[ f_z(z,t) \overline{V}_p(z,t) \overline{n}_p(z,t) \right]$$

$$= -P(z,t) \int_0^\infty \int_0^\infty \frac{\partial}{\partial n_p} \left[ V_p n_p f_{V_p,n_p,z}(V_p, n_p, z, t) \right] \, \mathrm{d}n_p \, \mathrm{d}V_p$$

$$+ P(z,t) \int_0^\infty \int_0^\infty V_p f_{V_p,n_p,z}(V_p, n_p, z, t) \, \mathrm{d}n_p \, \mathrm{d}V_p$$

$$- \lambda \int_0^\infty \int_0^\infty V_p n_p f_{V_p,n_p,z}(V_p, n_p, z, t) \, \mathrm{d}V_p \, \mathrm{d}n_p. \tag{17}$$

Assuming that $f_{V_p,n_p,z}(V_p, \infty, z, t) = 0$, the first double integral on the right side of Eq. (17) is equal to zero. We then write Eq. (17) as

$$\frac{\partial}{\partial t} \left[ f_z(z,t) \overline{V}_p(z,t) \overline{n}_p(z,t) \right]$$

$$= P(z,t) \int_0^\infty V_p \, \mathrm{d}V_p \int_0^\infty f_{V_p,n_p,z}(V_p, n_p, z, t) \, \mathrm{d}n_p$$

$$- \lambda \int_0^\infty \int_0^\infty V_p n_p f_{V_p,n_p,z}(V_p, n_p, z, t) \, \mathrm{d}V_p \, \mathrm{d}n_p$$

$$= P(z,t) \int_0^\infty V_p f_{V_p,z}(V_p, z, t) \, \mathrm{d}V_p$$

$$-\lambda \int_0^\infty \int_0^\infty V_p n_p f_{V_p,n_p,z}(V_p, n_p, z, t)\, \mathrm{d}V_p\, \mathrm{d}n_p. \tag{18}$$

Using $f_{V_p,z}(V_p, z, t) = f_z(z,t) f_{V_p|z}(V_p|z, t)$ then leads to the result that

$$\frac{\partial}{\partial t}\left[f_z(z,t)\overline{V}_p(z,t)\overline{n}_p(z,t)\right]$$

$$= P(z,t)f_z(z,t)\overline{V}_p(z,t) - \lambda f_z(z,t)\overline{V}_p(z,t)\overline{n}_p(z,t). \tag{19}$$

Thus, the production and decay terms to be added to Eq. (12) or Eq. (13) are given by the right side of Eq. (19).

### 3.3 Expected particle OSL ages

#### 3.3.1 Conservation of OSL age

In principle, the experimentally determined OSL burial age of a particle is independent of its size. In addition, as previously mentioned, quartz particles targeted for single-grain OSL analysis have a relatively narrow range of sizes (0.35 – 0.425 mm). For these reasons we may neglect particle volume in the following formulation.

Let $f_{A_p,z}(A_p, z, t)$ denote the joint probability density function of particle OSL ages $A_p$ and positions $z$. For particles with a given age $A_p$, and momentarily neglecting the production of age, the density $f_{A_p,z}(A_p, z, t)$ satisfies a Fokker-Planck equation of the form

$$\frac{\partial f_{A_p,z}(A_p, z, t)}{\partial t}$$

$$= -\frac{\partial}{\partial z}\left[w_p(z,t)f_{A_p,z}(A_p, z, t)\right.$$

$$\left. -\kappa_z(z,t)\frac{\partial f_{A_p,z}(A_p, z, t)}{\partial z}\right]. \tag{20}$$

We now define a conditional joint probability density function of ages $A_p$, namely,

$$f_{A_p|z}(A_p|z, t) = \frac{f_{A_p,z}(A_p, z, t)}{f_z(z,t)}. \tag{21}$$

With Eq. (20) and Eq. (21) in place, we multiply both by $A_p$, integrate with respect to $A_p$, then follow the same steps as presented in Section 3.2.1 above to give

$$\frac{\partial}{\partial t}\left[f_z(z,t)\overline{A}_p(z,t)\right] = -\frac{\partial}{\partial z}\left[w_p(z,t)f_z(z,t)\overline{A}_p(z,t)\right]$$

$$+\frac{\partial}{\partial z}\left(\kappa_z(z,t)\frac{\partial}{\partial z}\left[f_z(z,t)\overline{A}_p(z,t)\right]\right), \tag{22}$$

where $\overline{A}_p(z,t)$ is the expected particle OSL age. We now turn to the production term to be added to Eq. (22).

### 3.3.2 Production of OSL age

In the absence of advection and diffusion, the joint probability density $f_{A_p,z}(A_p,z,t)$ satisfies a statement of conservation of probability having the form of an advection equation with respect to the $A_p$ domain, namely,

$$\frac{\partial f_{A_p,z}(A_p,z,t)}{\partial t} = -S\frac{\partial f_{A_p,z}(A_p,z,t)}{\partial A_p}, \tag{23}$$

where the advective speed $S = \mathrm{d}A_p/\mathrm{d}t = 1$ is the rate at which particles accumulate OSL age. We then multiple Eq. (23) by $A_p$, integrate with respect to $A_p$, then follow the same steps as presented in Section 3.2.2 above to give

$$\frac{\partial}{\partial t}\left[f_z(z,t)\overline{A}_p(z,t)\right] = Sf_z(z,t). \tag{24}$$

Thus, the production term to be added to Eq. (22) is given by the right side of Eq. (24). We elaborate below in practical terms the relation between the rate $S$ and the radiation dose rate during particle motions within the soil.

### 3.3.3 Variance of OSL ages

Because of its significance for sampling of particles for OSL analysis, here we consider the variance of particle OSL ages. Let $f_{A_p,z}(A_p,z,t)$ denote the joint probability density function of ages $A_p$ and positions $z$. We now form the conditional probability density function,

$$f_{A_p|z}(A_p|z,t) = \frac{f_{A_p,z}(A_p,z,t)}{f_z(z,t)}. \tag{25}$$

Multiplying by $(A_p - \overline{A}_p)^2$, rearranging, and integrating with respect to $A_p$,

$$f_z(z,t)\int_0^\infty (A_p - \overline{A}_p)^2 f_{A_p|z}(A_p|z,t)\,\mathrm{d}A_p$$

$$= \int_0^\infty (A_p - \overline{A}_p)^2 f_{A_p,z}(A_p,z,t)\,\mathrm{d}A_p. \tag{26}$$

This yields

$$f_z(z,t)m_2(z,t) = \int_0^\infty (A_p - \overline{A}_p)^2 f_{A_p,z}(A_p,z,t)\,\mathrm{d}A_p, \tag{27}$$

where $m_2(z,t)$ denotes the variance of particle OSL ages.

In turn, multiplying Eq. (20) by $(A_p - \overline{A}_p)^2$ and integrating with respect to $A_p$ — recognizing that $\overline{A}_p$ is a function of position and time and therefore judiciously applying the product rule and Leibniz's rule — then leads to the conclusion that

$$\frac{\partial}{\partial t}[f_z(z,t)m_2(z,t)] = -\frac{\partial}{\partial z}[w_p(z,t)f_z(z,t)m_2(z,t)]$$

$$+\frac{\partial}{\partial z}\left(\kappa_z(z,t)\frac{\partial}{\partial z}[f_z(z,t)m_2(z,t)]\right)$$

$$+2\kappa_z(z,t)f_z(z,t)\left[\frac{\partial\overline{A}_p(z,t)}{\partial z}\right]^2. \tag{28}$$

Thus the variance $m_2(z,t)$ satisfies a Fokker-Planck equation with a source-like term involving the average age $\overline{A}_p(z,t)$. Because this term depends on the structure of $\overline{A}_p(z,t)$, it therefore is indirectly associated with the production of OSL age. However, it is straightforward to show that direct production of the variance $m_2(z,t)$ of OSL ages is zero.

### 3.4 Advection and diffusion

The Fokker-Planck equation is basically an advection-diffusion equation. But here we reemphasize that the $^{10}$Be concentration $n_p$ and the OSL age $A_p$ are intensive properties of individual particles, and the volume $V_p$ is an extensive property of individual particles. These quantities do not experience advection and diffusion as normally envisioned as occurring in a continuum. To be clear, the particles experience advection and diffusion, and the quantities $n_p$, $V_p$ and $A_p$ are merely carried with the particles.

With respect to a soil column with dimensions $XYh$, let us assume a great number $N$ of particles. Then the product $Nf_z(z,t)=c(z,t)$ may be interpreted as the expected number concentration of particles. That is, $c(z,t)XY\,\mathrm{d}z$ represents the expected number of particles within the volume $XY\,\mathrm{d}z$ between $z$ and $z+\mathrm{d}z$ at time $t$. We may then rewrite Eq. (1) as

$$\frac{\partial c(z,t)}{\partial t}=-\frac{\partial}{\partial z}\left[w_p(z,t)c(z,t)-\kappa_z(z,t)\frac{\partial c(z,t)}{\partial z}\right], \tag{29}$$

which looks like a familiar advection-diffusion equation.

Similarly, $Nf_z(z,t)\overline{V}_p(z,t)\overline{n}_p(z,t)=n(z,t)$ represents the expected number concentration of $^{10}$Be atoms, and $f_z(z,t)\overline{V}_p$ represents the volumetric particle concentration. We may then rewrite Eq. (13) as

$$\frac{\partial n(z,t)}{\partial t}=-\frac{\partial}{\partial z}\left[w_p(z,t)n(z,t)-\kappa_z(z,t)\frac{\partial n(z,t)}{\partial z}\right]$$

$$+P(z,t)-\lambda n(z,t), \tag{30}$$

where $P(z,t)$ now is interpreted as the production rate per unit volume of soil.

We write the product $Nf_z(z,t)\overline{A}_p(z,t)$ as $c(z,t)\overline{A}_p(z,t)$, noting that $c(z,t)$ now specifically refers to particles with finite OSL age $A_p$. To simplify the notation, we denote the first moment of particle OSL ages as $m_1(z,t)=\overline{A}_p(z,t)$. We may then rewrite Eq. (22) as

$$\frac{\partial}{\partial t}[c(z,t)m_1(z,t)]$$

$$=-\frac{\partial}{\partial z}\left(w_p(z,t)c(z,t)m_1(z,t)\right.$$

$$-\kappa_z(z,t)\frac{\partial}{\partial z}[c(z,t)m_1(z,t)]\Bigg) + Sc(z,t)\,. \tag{31}$$

For the variance $m_2(z,t)$,

$$\frac{\partial}{\partial t}[c(z,t)m_2(z,t)]$$

$$= -\frac{\partial}{\partial z}\Bigg(w_p(z,t)c(z,t)m_2(z,t)$$

$$-\kappa_z(z,t)\frac{\partial}{\partial z}[c(z,t)m_2(z,t)]\Bigg)$$

$$+2\kappa_z(z,t)c(z,t)\left[\frac{\partial m_1(z,t)}{\partial z}\right]^2\,. \tag{32}$$

## 4  The steady one-dimensional problem

We now turn to a benchmark situation inspired by the pioneering work of Lal (1991) and Lal and Chen (2005) concerning CRN profiles within rock, and within well mixed soils above rock, undergoing steady surface erosion. With reference to Figure 2, we imagine the idealized situation involving a one-dimensional vertical mean motion of particles through a soil column,

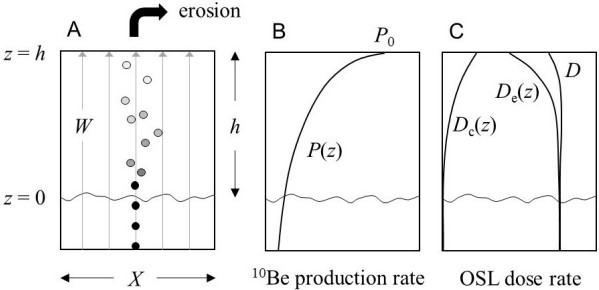

**Figure 2.** Schematic diagram of: (A) soil element with dimensions $XYh$. Particles move from the soil-saprolite interface ($z = 0$) into the element at a steady rate $W$ and are eroded from the surface ($z = h$). Particles experience a mean motion (gray arrows) with superimposed mixing motions. (B) *in situ* $^{10}$Be production rate $P(z)$. (C) idealized luminescence dose rate $D$ as the sum of the external rate $D_e(z)$ and the contribution from cosmic rays $D_c(z)$. Compare with Figure 1 in Mudd and Yoo (2010).

where steady surface erosion plus any chemical mass losses match the rate of soil production at the base of the column (e.g., Mudd and Yoo, 2010; Dixon and Riebe, 2014; Granger and Riebe, 2014). Although idealized, given that surface erosion rates generally are not steady (e.g., Small et al., 1997; Parker and Perg, 2005; Schaller et al., 2009), this benchmark nonetheless represents a valuable starting point for assessing actual conditions in field settings, including the possibility of a sudden change

in surface erosion (Granger and Riebe, 2014), and as a contrast for two-dimensional transport by soil creep (Small et al., 1999; Anderson, 2015; Furbish et al., 2018b). With respect to cosmogenic nuclides — $^{10}$Be in particular — previous formulations of this problem have focused on two end-member cases: absence of soil particle mixing, and the so-called "well mixed" case (or "complete" mixing) (e.g., Lal and Chen, 2005; Granger and Riebe, 2014), without reference to partial mixing or to the possible significance of the vertical structure of mixing, that is, whether particle mixing is uniform or depth dependent. This contrasts with the idea that soil distrubances and associated mixing likely involve a systematic depth dependence (Humphreys and Field, 1998; Cousins et al., 1999; Roering, 2004; Wilkinson et al., 2009). No analogous benchmark formulation exists for particle OSL ages.

We note that quartz enrichment (Small et al., 1999; Granger and Riebe, 2014) due to chemical weathering and mass loss may occur during any transient approach to steady conditions; but under steady conditions this enrichment does not impact the mechanical transport and mixing of quartz particles. In addition, we are for simplicity neglecting the vertical variation in soil bulk density that can occur with bioturbation (e.g., Furbish et al., 2009b, see Figure 4 therein).

In this steady problem, note that $w_p(z,t) = W$ and $\kappa_z(z,t) = \kappa_z(z)$. We consider two forms of the particle diffusivity $\kappa_z(z)$. In the first case we consider uniform mixing such that $\kappa_z(z) \to K_z$. In the second case we consider a linear variation in mixing such that $\kappa_z(z) = K_z z/h$. This represents the first-order structure of a depth dependency in mixing which, although currently not well constrained, appeals to the idea that disturbances leading to particle mixing systematically decline with depth (Humphreys and Field, 1998; Cousins et al., 1999; Roering, 2004; Wilkinson and and Humphreys, 2005; Wilkinson et al., 2009; Johnson et al., 2014). These two cases provide a straightforward contrast for considering how the form of $\kappa_z(z)$ might influence the profiles of $^{10}$Be concentration and particle OSL age. Following Furbish et al. (2018a, 2018b), we define a Péclet number as $Pe = Wh/K_z$. This provides a measure of the overall intensity of mixing. A large value of $Pe$ represents weak mixing, whereas a small value of $Pe$ represents strong mixing.

Following Furbish et al. (2018b), we assume that particles experience a constant radiation dose rate $D$ (Figure 2) during their motions within the soil column. Indeed, single-grain OSL systematics require assuming a constant natural dose rate in order to calculate a burial age $A_p$ from the measured particle luminescence and a regeneration curve created by subjecting the particle to varying experimental "equivalent dose" values (Duller, 2008). But the natural dose rate that a particle experiences may vary with its position, and therefore with time, as the particle moves up and down within the soil column. This means that a particle will yield a luminescence signal, and thus an OSL age, that depends on its history of exposure to different dose rates; but this particle history cannot be inferred in the experimental determination of its OSL age.

With respect to the source $S = \mathrm{d}A/\mathrm{d}t = 1$ of particle OSL aging in Eq. (31), we are essentially assuming, as described above, that particles experience a uniform radiation dose rate during their motions within the soil column. Namely, assuming homogeneous soil material and moisture content, the external dose rate $D_e(z)$ supplied by the radioactive decay of elements within the surrounding soil is uniform below ∼30 cm (or less (Aitken et al., 1985)) and declines toward the soil surface because of the incomplete gamma dose field at shallow depths (Figure 2C). The dose rate $D_c(z)$ due to cosmic rays (varying with latitude and altitude) declines nonlinearly below the soil surface (Prescott and Hutton, 1988, 1994). The total dose rate $D(z)$ equals the sum of the external and cosmic rates. In general, the cosmic contribution tends to offset the decline of the

external rate. But this depends on the relative magnitudes of these two contributions, where the magnitude of the external rate is determined by the mineral content of the soil, and the associated concentration of radioactive elements.

If the magnitude of the cosmic dose rate is similar to that of the external dose rate near the soil surface, then the total dose rate is approximately uniform (Figure 2C). If, however, the cosmic rate does not fully offset the decrease in the external rate, we nonetheless suggest that the assumption of a uniform dose rate is a reasonable starting point for comparison with deviations in OSL age profiles that might be expected from a nonuniform dose rate, particularly under conditions of moderate to strong particle mixing, whose effects likely mask spatial variations in the total dose rate (e.g., Furbish et al., 2018b). That is, this is a parsimonious assumption — that the effects of mixing of ages outweigh any consequence of a nonuniform dose field. Previous studies using luminescence to examine soil mixing show relatively uniform total dose rates (e.g., Heimsath et al., 2002, Johnson et al., 2014).

In order to present our results below in a manner that highlights the effects of differences in the intensity and depth dependence of particle mixing, it is convenient to define the following dimensionless quantities denoted by circumflexes:

$$\hat{z} = \frac{z}{h}, \quad \hat{c}(\hat{z}) = \frac{c(z)}{c(h)}, \quad \hat{n}(\hat{z}) = \frac{n(z)}{n(h)}$$

and $\quad \hat{m}_j(\hat{z}) = \left(\frac{W}{h}\right)^j m_j(z).$ \hfill (33)

Here, $\hat{z}$ denotes the dimensionless height within the soil column above the soil-saprolite interface, $\hat{n}(\hat{z})$ denotes the dimensionless concentration of $^{10}$Be atoms relative to the concentration at the soil surface, $\hat{c}(\hat{z})$ denotes the dimensionless number concentration of particles with finite OSL ages relative to the concentration at the soil surface, and $\hat{m}_j(\hat{z})$ denotes the $j$th moment ($j = 1, 2$) of OSL ages relative to the mean residence time, $h/W$, of target quartz particles.

The analytical results presented in the next two sections involving $\hat{n}(\hat{z})$ and $\hat{m}_j(\hat{z})$ are derived in the appendixes of this paper. As described therein, each of the statements of conservation above must satisfy specific boundary conditions that depend on uniform versus nonuniform particle mixing. Here are key constraints. The $^{10}$Be flux across the soil surface equals the flux into the soil column across the soil-saprolite interface plus the total production of $^{10}$Be within the column. The flux of particles with finite OSL age across any surface normal to $z$ is zero, and the concentration of these particles at the surface is equal to the concentration of OSL sensitive particles entering the base of the column, although these take on finite OSL ages only after they reach the surface and are bleached. The expected OSL age at the soil surface is zero, and a diffusive flux of age across the surface matches the total production of age within the column. Particles with finite OSL age cannot be imported to the soil column. We defer commenting on the results presented next until we present our numerical simulations in Section 5.

### 4.1 Expected $^{10}$Be concentrations

Assuming that $^{10}$Be production is due to spallation (e.g., Gosse and Phillips, 2001), the production rate $P(z)$ in Eq. (30) is (Lal, 1991; Small et al., 1999)

$$P(z) = P_0 e^{-(h-z)/l_s},$$ \hfill (34)

where $P_0$ is the $^{10}$Be production rate at the surface and $l_s$ is the $e$-folding attenuation length of the soil. Neglecting the decay of $^{10}$Be with its half-life of $\sim 10^6$ years, then for steady conditions Eq. (30) becomes

$$\frac{\mathrm{d}}{\mathrm{d}z}\left[Wn(z) - \kappa_z(z)\frac{\mathrm{d}n(z)}{\mathrm{d}z}\right] = P_0 e^{-(h-z)/l_s}\,. \tag{35}$$

For uniform mixing with $\kappa_z(z) \to K_z$, the solution of Eq. (35) is (Appendix B)

$$\hat{n}(\hat{z}) = e^{-Pe(1-\hat{z})}$$

$$+\frac{Pe_{l_s}}{Pe_{l_s} - 1}\left[e^{-h(1-\hat{z})/l_s} - e^{-Pe(1-\hat{z})}\right]\,, \tag{36}$$

where $Pe_{l_s} = Wl_s/K_z$ is a secondary Péclet number. In turn, for nonuniform mixing with $\kappa_z(z) = K_z z/h$, the solution of Eq. (35) is

$$\hat{n}(\hat{z}) = \hat{z}^{Pe} + Pe\left[\left(-\frac{h\hat{z}}{l_s}\right)^{Pe}\Gamma\left(-Pe, -\frac{h\hat{z}}{l_s}\right)\right.$$

$$\left.-\left(-\frac{h}{l_s}\right)^{Pe}\Gamma\left(-Pe, -\frac{h}{l_s}\right)\hat{z}^{Pe}\right]e^{-h/l_s}\,, \tag{37}$$

where $\Gamma$ denotes the incomplete gamma function. Note that Eq. (37) has real and imaginary parts. Only the real part is physically meaningful in this problem.

## 4.2 Expected particle OSL ages

Recall that the number concentration $c(z,t)$ in Eq. (31) specifically refers to particles with finite OSL age. For steady conditions Eq. (1) becomes

$$\frac{\mathrm{d}}{\mathrm{d}z}\left[Wc(z) - \kappa_z(z)\frac{\mathrm{d}c(z)}{\mathrm{d}z}\right] = 0\,. \tag{38}$$

Using this, Eq. (31) is simplified to

$$\frac{\mathrm{d}}{\mathrm{d}z}\left[-\kappa_z(z)c(z)\frac{\mathrm{d}m_1(z)}{\mathrm{d}z}\right] = Sc(z)\,. \tag{39}$$

For the variance $m_2(z)$,

$$\frac{\mathrm{d}}{\mathrm{d}z}\left[-\kappa_z(z)c(z)\frac{\mathrm{d}m_2(z)}{\mathrm{d}z}\right]$$

$$-2\kappa_z(z)c(z)\left[\frac{\mathrm{d}m_1(z)}{\mathrm{d}z}\right]^2 = 0\,. \tag{40}$$

Note that Eq. (39) and Eq. (40) involve only diffusion, not advection. Advection and diffusion of particles possessing finite OSL ages involve the transport and mixing of OSL ages, thus influencing the age moments. But because upward advection of these particles is balanced by downward diffusion under steady conditions, this balance sets the OSL age structure wherein diffusion maintains the steady, finite values of the age moments in the presence of production of OSL age.

For uniform mixing with $\kappa_z(z) \to K_z$, the solution of Eq. (38) is (Appendix C)

$$\hat{c}(\hat{z}) = e^{-Pe(1-\hat{z})}. \tag{41}$$

For nonuniform mixing with $\kappa_z(z) = K_z z/h$, the solution of Eq. (38) is

$$\hat{c}(\hat{z}) = \hat{z}^{Pe}. \tag{42}$$

In turn, using these results for $\hat{c}(\hat{z})$, for uniform mixing the solution of Eq. (39) is (Appendix D)

$$\hat{m}_1(\hat{z}) = S(1 - \hat{z}) + \frac{Se^{-Pe}}{Pe}\left[1 - e^{Pe(1-\hat{z})}\right], \tag{43}$$

and for nonuniform mixing the solution of Eq. (39) is (Appendix D)

$$\hat{m}_1(\hat{z}) = \frac{SPe}{1 + Pe}(1 - \hat{z}). \tag{44}$$

For the variance $\hat{m}_2$ the solution of Eq. (40) for uniform mixing is (Appendix E)

$$\hat{m}_2(\hat{z}) = \frac{2S^2}{Pe}(1 - \hat{z})$$

$$+ \frac{4S^2}{Pe^2}\left[(1 + Pe)e^{-Pe} - (1 + Pe\hat{z})e^{-Pe\hat{z}}\right]$$

$$+ \frac{S^2}{Pe^2}\left(e^{-2Pe} - e^{-2Pe\hat{z}}\right). \tag{45}$$

and for nonuniform mixing the solution of Eq. (40) is (Appendix E)

$$\hat{m}_2(\hat{z}) = \frac{S^2 Pe^2}{(2 + Pe)(1 + Pe)^2}\left(1 - \hat{z}^2\right). \tag{46}$$

Also for reference below, the column-averaged particle OSL age $\hat{M}_1$ within the soil is

$$\hat{M}_1 = \int_0^1 \hat{c}(\hat{z})\hat{m}_1(\hat{z})\,d\hat{z}. \tag{47}$$

For uniform mixing, Eq. (41) and Eq. (43) lead to

$$\hat{M}_1 = \frac{S}{Pe^2}\left(1 - e^{-2Pe}\right) - \frac{2Se^{-Pe}}{Pe}. \tag{48}$$

For nonuniform mixing, Eq. (42) and Eq. (44) lead to

$$\hat{M}_1 = \frac{SPe}{1 + Pe}\left(\frac{1}{1 + Pe} - \frac{1}{2 + Pe}\right). \tag{49}$$

We comment further on the results above after presenting our numerical simulations.

# 5  Numerical simulations

We now turn to numerical simulations of particles undergoing random-walk motions within the soil column, during which they accumulate [10]Be atoms within the production field, and undergo OSL "aging" following their most recent encounters with the soil surface. These simulations have two purposes.

First, the random-walk motions implied by the probabilistic formulations above are in principle straightforward to implement numerically, and it is important to demonstrate that such computational results match the analytical results presented. In doing this, the simulations reveal important information that is not readily apparent in the analytical results. This includes an illustration of the variability in [10]Be concentrations and OSL ages of individual particles, in contrast to expected values at positions $z$, with important implications for interpreting field-based measurements, and the nature of the terms in Eq. (14) and Eq. (23) describing production of [10]Be atoms and particle OSL age.

Second, numerical simulations of particle motions within soils offer important opportunities to examine phenomena that cannot readily be treated analytically, for example, effects of particle residence times on mineral weathering, or effects of a nonuniform radiation dose rate. So, spinning our first objective around, any numerical simulation of random walk motions must be able to correctly reproduce benchmark (analytical) solutions before being applied to more complex situations, for example, two-dimensional motions and unsteady conditions. The simulations presented here highlight important aspects involved.

Following Furbish et al. (2018a, 2018b), we adopt a straightforward Eulerian-Lagrangian algorithm to simulate particle motions in a mass conserving manner. Particles are numerically introduced to the base of the soil column ($z = 0$), then undergo a mean upward motion equal to $W$ with superimposed Gaussian fluctuations. For uniform mixing the particle diffusivity is set as $\kappa_z = K_z$, and the random walk becomes

$$z(t + \Delta t) = z(t) + W\Delta t + R_z(a), \tag{50}$$

where $\Delta t$ denotes the time step and $R_z(a)$ is a Gaussian random variable with argument $a = (2K_z\Delta t)^{1/2}$. For nonuniform mixing with $\kappa(z) = K_z z/h$, the random walk becomes

$$z(t + \Delta t) = z(t) + W\Delta t + R_z(a) + \kappa'_z \Delta t, \tag{51}$$

with argument $a = [2\kappa_z(z + 0.5\kappa'_z \Delta t)\Delta t]^{1/2}$, where $\kappa'_z = \partial \kappa_z(z)/\partial z$. This yields a mass conserving behavior, that is, one that prevents particles from unrealistically drifting from sites with high particle diffusivity to sites with low diffusivity. Moreover, this algorithm has been shown to work for variations in diffusivity that are not linear (e.g., Legg and Raupach, 1982; Hunter et al., 1993; Visser, 1997). The theoretical basis of Eq. (51) and its relation to the Fokker-Planck equation are covered in these references and in Appendix G of Furbish et al. (2018a).

Each particle accumulates [10]Be atoms as a function of its local position $z$, and it accumulates a numerical OSL age from the time of its last encounter with the soil surface. We spin up each simulation to a steady-state condition, where the rate at which particles exit the soil column is equal to the rate at which they are introduced at the base, and particles within the column are distributed uniformly over the thickness $h$. The total spin-up time involves at least four $e$-folding residence times $h/W$. At

steady state, the total number of particles within the column is $N_T \approx N_c(h/W)/\Delta t$, where $N_c$ is the number of particles in the cohort introduced at each time step. We use a minimum of $N_T \approx 10{,}000$ for the $^{10}$Be simulations.

The lower boundary ($z = 0$) is treated as a reflecting boundary. For each particle reaching the upper boundary ($z = h$), it either may leave the column with a specified probability that ensures global particle conservation, or it is reflected. In the case of particle OSL ages, the numerical age of an individual particle is set to zero if it is reflected at $z = h$. The effect of this is to correctly mimic the boundary condition in the formulation above, that $\hat{m}_j = 0$ at $\hat{z} = 1$. In actuality, however, bleaching of particles can occur just below the soil surface with light penetration (to a few particle diameters) and with heating from fires at the surface (Wilkinson and Humphreys, 2005; Duller, 2008), such that actual values $\hat{m}_j = 0$ occur below the soil surface.

All simulated $N_T$ particles at steady state possess a $^{10}$Be value. But only a proportion of these $N_T$ particles possess finite OSL ages at steady state, as not all of them reach the surface to subsequently take on finite OSL ages. We cannot know this proportion *a priori*. Thus, it is important to insist on global particle conservation in the simulations, involving verification of a specified $N_T$ together with a uniform distribution of particle positions $z$. In addition, we increase $N_T$ (up to 20,000) and the total spin-up time (up to six residence times $h/W$) for the OSL simulations to ensure that a sufficiently large number of particles is included in our calculations of expected values. However, this is not entirely possible with large Péclet number $Pe$, as described below.

## 5.1  $^{10}$Be concentrations

The simulated, expected $^{10}$Be concentrations closely match the theoretical results for different values of the Péclet number $Pe$ involving both uniform mixing (Figure 3) and nonuniform mixing (Figure 4). These profiles show that with weak mixing (large $Pe$), the expected concentration approaches the original exponential solution provided by Lal (1991). With strong mixing (small $Pe$), the expected concentration becomes increasingly uniform over the soil column, approaching the concentration at the soil surface. With uniform mixing (Figure 3), the concentration $\hat{n}(0)$ may be finite, as diffusion effectively moves particles downward to the soil-saprolite interface. With nonuniform mixing (Figure 4), the concentration $\hat{n}(0)$ is anchored by the value within the saprolite, as diffusion weakens downward then vanishes at the soil-saprolite interface.

With both uniform and nonuniform mixing, the distribution $f_{\hat{n}_p}(\hat{n}_p, \hat{z})$ of $^{10}$Be concentrations $\hat{n}_p$ of individual particles within any small interval $\mathrm{d}\hat{z}$ systematically varies with vertical position and the Péclet number $Pe$ (Figure 5). Notably, this distribution at any $\hat{z}$ is approximately symmetrical about the expected value for large $Pe$, and becomes increasingly skewed with decreasing $Pe$. The expected concentration $\hat{n}(\hat{z})$ at small $Pe$ thus is strongly influenced by the tail of this distribution, that is, by particles possessing concentrations much larger than the modal concentration.

## 5.2  Particle OSL ages

The simulated, expected OSL ages closely match the theoretical results for different values of the Péclet number $Pe$ involving both uniform mixing (Figure 6) and nonuniform mixing (Figure 7), where we note that the simulations yield meaningful results only near the surface for large Péclet number $Pe$. (Because the concentration $\hat{c}(\hat{z})$ of particles with finite OSL ages declines rapidly with depth for large $Pe$ (Appendix C), achieving reasonable numerical values of the expected age $\hat{m}_1(\hat{z})$ over the

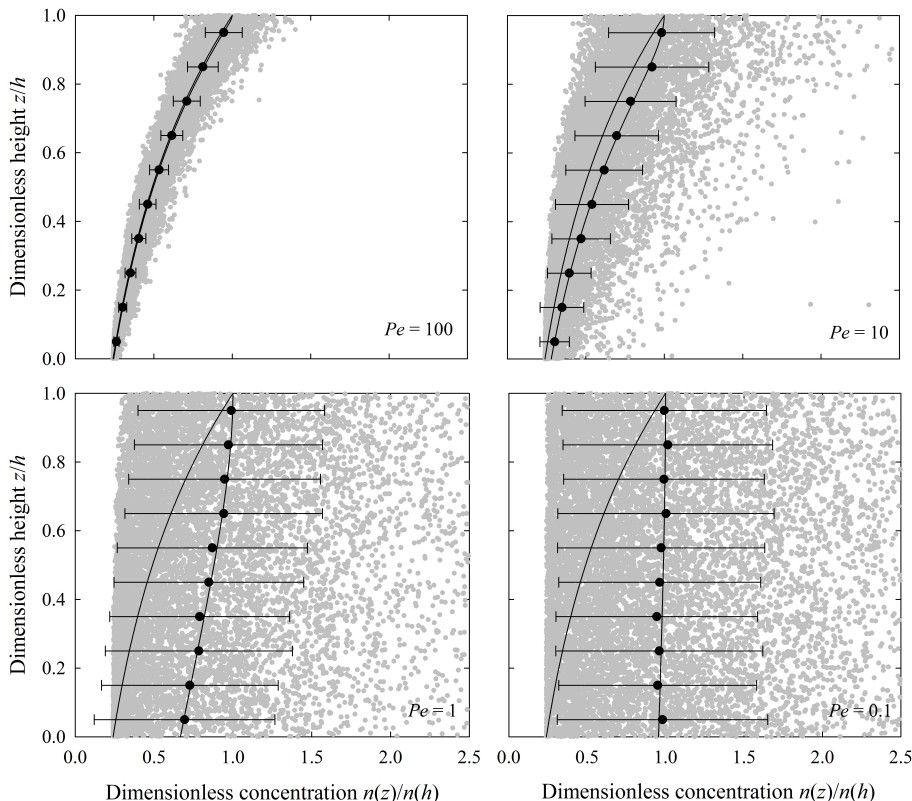

**Figure 3.** Plot of dimensionless $^{10}$Be concentration $\hat{n} = n(z)/n(h)$ versus dimensionless height $\hat{z} = z/h$ showing simulated particle concentrations $\hat{n}_p$ (gray dots) for $Pe = 100, 10, 1, 0.1$, and estimates of expected concentrations $\hat{n}$ averaged within $0.1h$ intervals (black circles) with one standard deviation bars. Simulations represent uniform mixing with $\kappa_z = K_z$. Right solid line is the theoretical result, and left solid line represents the absence of mixing.

entire soil thickness would require unreasonably large computational memory and time.) These profiles show that with weak mixing (large $Pe$), the expected particle OSL age increases linearly, or approximately linearly, with depth. With strong mixing (small $Pe$), the expected age becomes increasingly uniform and close to zero over the soil column. With uniform mixing, the diffusive flux of age must vanish at the soil-saprolite interface, so with finite diffusivity $K_z$, the slope $\mathrm{d}\hat{m}_1/\mathrm{d}\hat{z}|_{\hat{z}=0} = 0$. With nonuniform mixing, the diffusive flux of age likewise vanishes at the soil-saprolite interface as the diffusivity goes to zero. But the magnitude of the slope $\mathrm{d}\hat{m}_1/\mathrm{d}\hat{z}$ is finite near this interface in order to compensate the decreasing diffusivity.

With both uniform and nonuniform mixing, the distribution $f_{\hat{A}_p}(\hat{A}_p, \hat{z})$ of particle OSL ages within any small interval $\mathrm{d}\hat{z}$ mostly is highly skewed (Figure 8). This skew increases with decreasing Péclet number $Pe$. Particularly with nonuniform mixing, the expected OSL age $\hat{m}_1(\hat{z})$ thus is strongly influenced by the tail of this distribution, that is, by particles possessing finite ages much larger than the modal age.

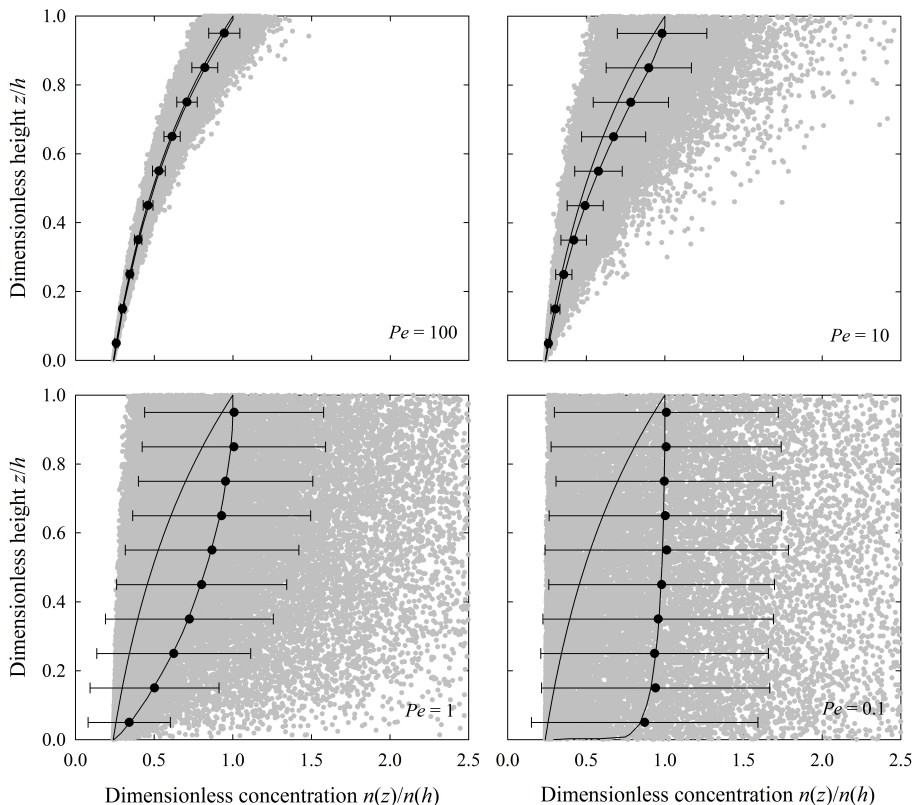

**Figure 4.** Plot of dimensionless $^{10}$Be concentration $\hat{n} = n(z)/n(h)$ versus dimensionless height $\hat{z} = z/h$ showing simulated particle concentrations $\hat{n}_p$ (gray dots) for $Pe = 100, 10, 1, 0.1$, and estimates of expected concentrations $\hat{n}$ averaged within $0.1h$ intervals (black circles) with one standard deviation bars. Simulations represent nonuniform mixing with $\kappa_z = K_z z/h$. Right solid line is the theoretical result, and left solid line represents the absence of mixing.

The simulated second moment $\hat{m}_2(\hat{z})$ of OSL ages reasonably matches the theoretical results for different values of the Péclet number $Pe$ for both uniform and nonuniform mixing. Focusing on the example of $Pe = 1$ (Figure 9), the variance $\hat{m}_2(\hat{z})$ rapidly increases with depth from zero at the soil surface, then becomes relatively uniform with increasing depth. With both uniform and nonuniform mixing, the variance at any position $\hat{z}$ generally decreases with decreasing $Pe$ (Figures 6 and 7). We note that, whereas in any individual simulation the numerical estimates of the expected OSL ages $\hat{m}_1(\hat{z})$ closely match the theoretical values with large $N_T$ for small $Pe$ (Figures 6 and 7) — a consequence of the central limit theorem — numerical estimates of the variance $\hat{m}_2(\hat{z})$ may fluctuate about the theoretical values from one simulation to the next (Figure 9).

The simulations suggest that particle OSL ages within the entire soil column are distributed approximately exponentially for both uniform and nonuniform mixing (Figure 10), where the column-averaged age $\hat{M}_1$ varies systematically with the Péclet number $Pe$. Interestingly, based on Eq. (48) and Eq. (49), the average $\hat{M}_1$ increases from zero at $Pe \to 0$, reaches a maximum of $\hat{M}_1 \sim 0.1$ near $Pe \sim 1$, then declines again with increasing $Pe$ (Figure 11), consistent with the simulations (Figure 10). For

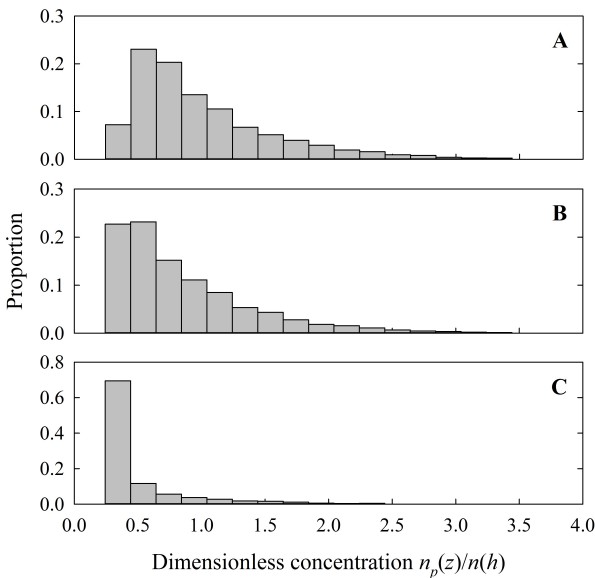

**Figure 5.** Example histograms representing the distribution $f_{\hat{n}_p}(\hat{n}_p, \hat{z})$ using simulated values of $\hat{n}_p$ from Figure 4 ($Pe = 1$) over the intervals (A) $0.9\hat{z} - 1.0\hat{z}$, (B) $0.5\hat{z} - 0.6\hat{z}$ and (C) $0.1\hat{z} - 0.2\hat{z}$. Analogous histograms associated with uniform mixing show a similar structure.

$Pe \to 0$, small values of $\hat{M}_1$ reflect the idealized condition of complete mixing, where particles that reach the soil surface and are bleached and then move downward rather than being eroded, nonetheless frequently return to the soil surface due to strong mixing. For large $Pe$, small values of $\hat{M}_1$ reflect that particles with finite OSL age tend to remain near the soil surface due to the strong effect of upward advection, and thus frequently return to it, many exiting by erosion before accumulating large ages.

5 Relatively large values of $\hat{M}_1$ at intermediate $Pe$ reflect the effects of an approximate balance between upward advection and downward diffusion of particles with finite OSL age, such that particles return to the soil surface less frequency. We emphasize that the maximum value of $\hat{M}_1$ is a fraction of the mean residence time $h/W$.

## 6 Discussion and Conclusions

### 6.1 Implications of rarefied transport conditions

10 We emphasize that, in contrast to continuum formulations of advection and diffusion of material (e.g., mass) measured as an intensive quantity (e.g., concentration) of the continuum, the extensive and intensive particle properties $V_p$, $n_p$ and $A_p$ "belong" to the particles, not to the bulk soil. For this reason, a formulation of advection and diffusion of [10]Be concentrations and expected particle OSL ages based on the Fokker-Plank equation provides a satisfactory way to parse the behavior of the particle-centric quantities $V_p$, $n_p$ and $A_p$. In the case of [10]Be, the formulation describes the behavior of the expected value 15 of individual particle concentrations at a position $z$. When this is combined with the expected particle volume and number

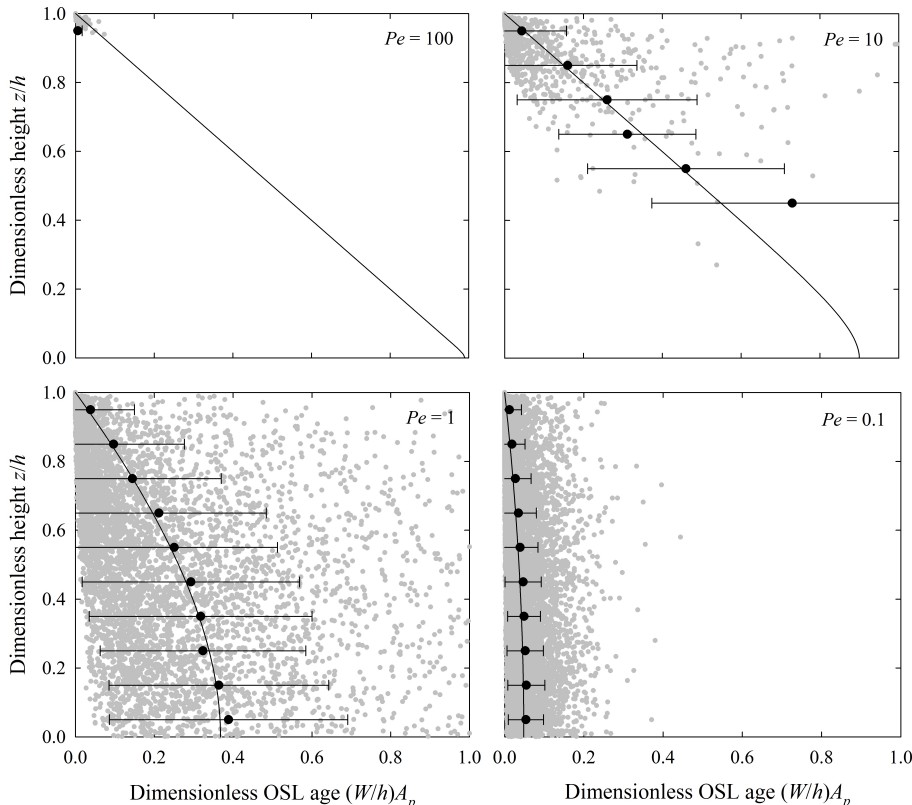

**Figure 6.** Plot of dimensionless OSL age $\hat{A}_p = (W/h)A_p$ versus dimensionless height $\hat{z} = z/h$ showing simulated particle ages $\hat{A}_p$ (gray dots) for $Pe = 100, 10, 1, 0.1$, and estimates of expected values $\hat{m}_1$ averaged within $0.1h$ intervals (black circles) with one standard deviation bars. Simulations represent uniform mixing with $\kappa_z = K_z$. Solid line is the theoretical result.

concentration, the expected $^{10}$Be concentration $n(z,t)$ then may be considered an intensive property of the soil at position $z$. As a consequence, the expected concentration $n(z,t)$ satisfies what looks like an ordinary advection-diffusion equation with production and decay terms — although this does not necessarily imply a continuum behavior (Section 3.1).

In the case of particle OSL ages, the formulation similarly describes the behavior of the expected value (and the variance) of
5   individual particle OSL ages at a position $z$. By definition, our interest is in this expected particle OSL age, as this is what is determined from single-grain OSL measurements. It therefore does not make sense to define OSL age as an intensive property of the soil by combining the expected particle OSL age with the expected particle number concentration (resulting in a "total" OSL age at position $z$). Moreover, by maintaining this distinction, the formulation reveals that the expected particle OSL age (and the variance) satisfy a diffusion-like equation according to Eq. (39) and Eq. (40), not an advection-diffusion equation. This
10   is in contrast to the idea that the "age" of a fluid parcel moving through a continuum domain satisfies an advection-diffusion equation with a production term equal to unity, as described in oceanographic and hydrological applications (England, 1995; Goode, 1996). This is important because, unlike a continuum material, the expected number concentration $c(z,t)$ of particles

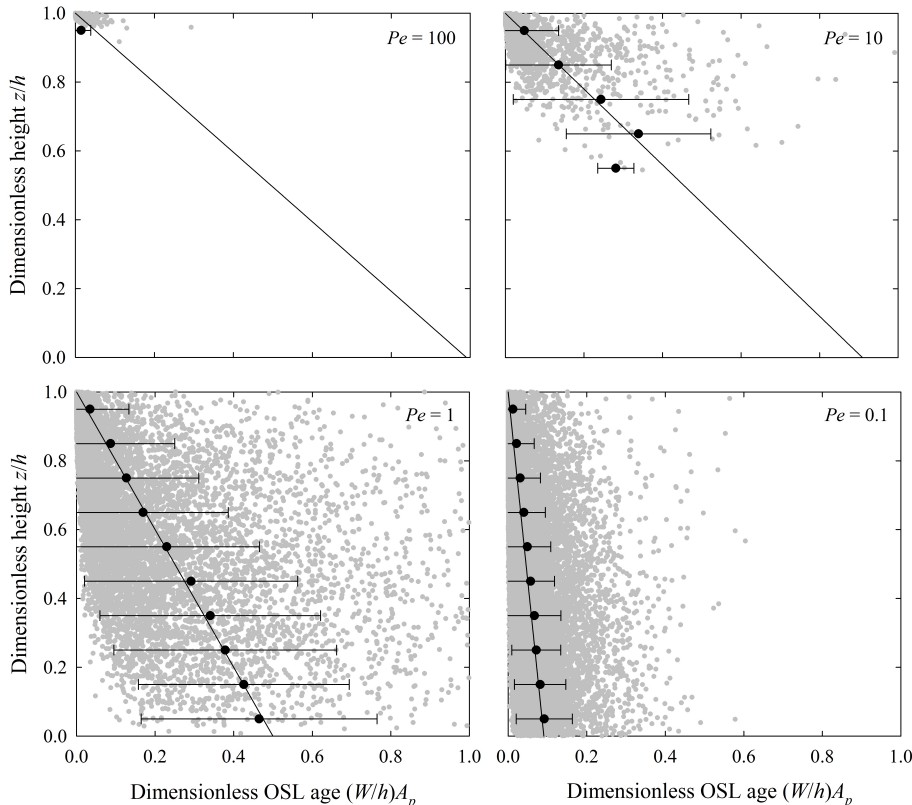

**Figure 7.** Plot of dimensionless OSL age $\hat{A}_p = (W/h)A_p$ versus dimensionless height $\hat{z} = z/h$ showing simulated particle ages $\hat{A}_p$ (gray dots) for $Pe = 100, 10, 1, 0.1$, and estimates of expected values $\hat{m}_1$ averaged within $0.1h$ intervals (black circles) with one standard deviation bars. Simulations represent nonuniform mixing with $\kappa_z = K_z z/h$. Solid line is the theoretical result.

possessing a finite OSL age generally is not uniform over $z$ (Appendix C). That is, this concentration does not mimic a uniform continuum domain within which particle OSL age is transported.

An essential lesson is this. When the quantity of interest can be expressed as a total value within an interval $dz$, as with the total number of $^{10}$Be atoms, then this quantity may be treated as an intensive property of the bulk soil. When the quantity of

5   interest is an expected value within $dz$, as with the moments $m_j(z)$ of particle OSL age, then this quantity cannot be expressed as an intensive property of the bulk soil, and its behavior must be coupled with that of the expected concentration $c(z)$ of the particles possessing the property. Similar quantities include, for example, particle size (in relation to descriptions of vertical sorting (Campforts et al., 2016)) and particle age as measured from the time of entry into the mechanically active soil column (in relation to studies of particle weathering (White and Brantley, 2003; Mudd and Furbish, 2006; Almond et al., 2007; Anderson

10  et al., 2007; Yoo and Mudd, 2008; Mudd and Yoo, 2010; Ferrier et al., 2016)). In contrast, there is a growing interest in the use and interpretation of the total OSL intensity of bulk soil samples as measured by portable OSL readers (Muñoz-Salinas et al., 2010; Sanderson and Murphy, 2010; Stang et al., 2012; Munyikwa and Brown, 2014; Gray et al., 2017; Gray, 2018; Porat

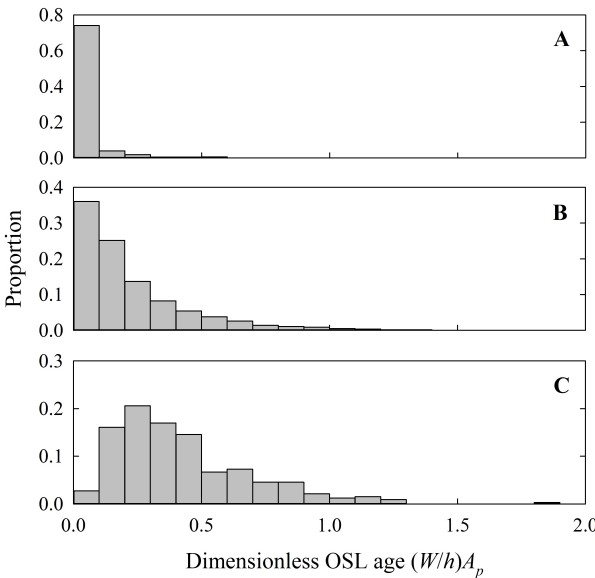

**Figure 8.** Example histograms representing the distribution $f_{\hat{A}_p}(\hat{A}_p, \hat{z})$ using simulated values of $\hat{A}_p$ from Figure 7 ($Pe = 1$) over the intervals (A) $0.9\hat{z} - 1.0\hat{z}$, (B) $0.5\hat{z} - 0.6\hat{z}$ and (C) $0.1\hat{z} - 0.2\hat{z}$. Analogous histograms associated with uniform mixing show a similar structure.

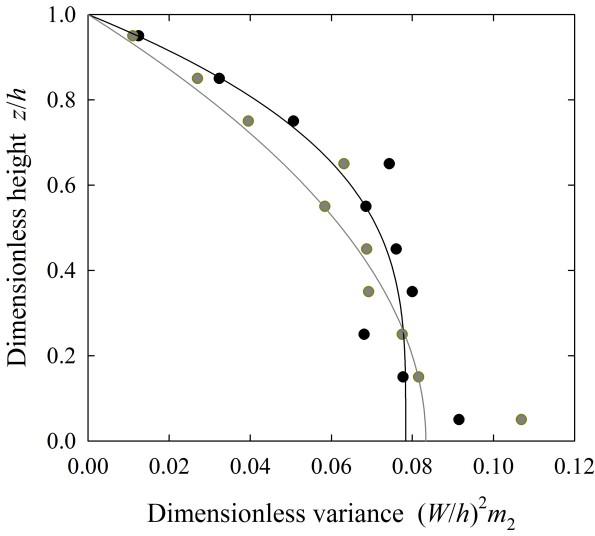

**Figure 9.** Plot of dimensionless variance $\hat{m}_2 = (W/h)^2 m_2$ versus dimensionless height $\hat{z} = z/h$ showing values obtained from simulations ($Pe = 1$) for uniform mixing (black circles) and nonuniform mixing (gray circles) compared with theoretical values (black and gray lines).

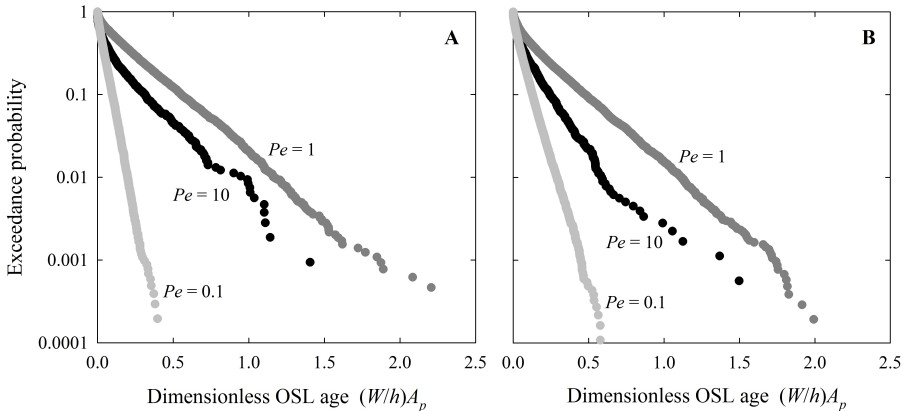

**Figure 10.** Exceedance probability plots of dimensionless particle OSL age $\hat{A}_p = (W/h)A_p$ for (A) uniform mixing and (B) nonuniform mixing for Péclet numbers $Pe = 10$, 1 and 0.1.

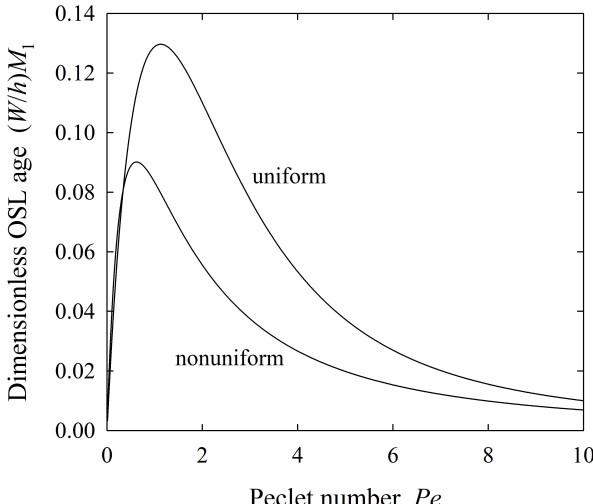

**Figure 11.** Plot of dimensionless column-averaged OSL age $\hat{M}_1 = (W/h)M_1$ versus Péclet number $Pe = Wh/K_z$ for uniform and nonuniform mixing.

et al., 2018). The luminescence intensities of individual particles — decidedly a random variable (Gray, 2018) — contribute to the total measured intensity. Thus, because the quantity of interest is the total intensity rather than expected moments of individual particle intensities, the total intensity can be formulated as being an intensive property of the bulk soil (Gray et al., 2017; Gray, 2018).

5    Throughout we have emphasized that $^{10}$Be concentrations and OSL ages are to be considered expected values. Moreover, this expectation is defined with respect to an interval $z$ to $z + \mathrm{d}z$ in a soil element with finite areal dimension $XY$, and it

formally is an ensemble average, rather than the expected value associated with an individual realization. The significance of this bears on the practical issue of sampling soil material for measurements of $^{10}$Be and particle OSL age, in view of the fact that disturbance driven particle motions in soils are patchy and intermittent at many scales, where most particles are at rest most of the time. Namely, vertical profiles of soil properties measured in an individual soil pit (where $XY$ is on the order of $1 \times 1$ m) reflect a "snapshot" of possible conditions (Furbish et al., 2009b). This snapshot represents the recent history of transport and mixing, one that is much shorter than the typical soil particle residence time, $W/h$.

We cannot avoid this issue of legacy (or "inheritance"), namely, the likelihood that what is being measured reflects only the recent history of transport and mixing as opposed to conditions consistent with an imagined behavior averaged over longer timescales, as represented by the expected profiles in Figures 3, 4, 6 and 7 above. In the case of measured profiles of $^{10}$Be concentrations and particle OSL ages, this has two parts. Consider a profile that reflects an expected steady-state condition (Figures 3, 4, 6 or 7). Disturbances that contribute to the mixing motions consistent with the profile may occur at different length scales and with different frequencies, where large disturbances may involve coherent motions whose effects are more akin to stirring than mixing, thus momentarily producing irregularities about the expected state. To the extent that mixing is adequately characterized as being diffusive, then we may define a relaxation timescale as $T = r^2/\kappa \sim r^2/\langle r^2 \rangle f$, where now the mixing motion $r$ is used as a measure of the length scale of disturbance, $f$ denotes a characteristic frequency of disturbance, and the angle brackets denote ensemble averaging. With $r^2$ in the numerator and $\langle r^2 \rangle$ in the denominator, this expression highlights the duality of disturbances, that these provide mixing motions, yet this mixing is responsible for diffusive smoothing of disturbance produced irregularities about the expected profile state. This is in marked contrast to, say, classic molecular diffusion, where molecular motions smooth irregularities, but are not the source of disturbances to the expected state. Thus, for a given ensemble averaged disturbance magnitude $\langle r^2 \rangle^{1/2}$, the relaxation time $T$ goes with the square of the scale of disturbance and inversely with the characteristic frequency of disturbances. For a given frequency $f$, effects of big disturbances tend to persist whereas effects of small disturbances do not. In either case, this persistence decreases with increasing disturbance frequency $f$ (i.e., decreasing Péclet number $Pe$).

We now take the ensemble average of relaxation timescales $T$ over all disturbance length scales, namely, $\langle T \rangle \sim 1/f$. This indicates that the overall relaxation in response to a range of disturbance scales goes simply with the reciprocal of the disturbance frequency $f$. Thus, regardless of the mixture of disturbance scales involved, the disturbance frequency has a dominant role in setting the relaxation timescale. Then, for example, if disturbances and mixing motions are consistently small and relatively uniform in comparison to the size of the soil pit (and the size of individual soil samples), and if the frequency of the disturbances is sufficiently high, then one might anticipate observing at any instant only small variations about the expected steady-state profile. If, however, disturbances are infrequent and patchy at the scale of the soil pit or larger, then one might anticipate a greater likelihood of observing conditions unlike the expected profile. Conversely, frequent and spatially uniform large disturbances likely would lead to wholesale homogenization of tracer particles.

This points to the need to avoid over-interpreting the forms of profiles from individual soil pits in terms of what these forms might reflect about the vertical structure of mixing (e.g., uniform versus depth dependent mixing). Unfortunately, this issue is exacerbated by the reality that digging soil pits and sampling for $^{10}$Be concentrations and particle OSL ages is quite laborious,

and subsequent analytical analyses are prohibitively expensive. In addition, in choosing soil pit sites, we often avoid sites with evidence of recent disturbance. On the one hand, this strategy may obviate the sampling of conditions that likely deviate from averaged conditions; but on the other hand, it neglects observing profile irregularities that reflect the full range of disturbance scales. Connecting sampling strategies (e.g., involving multiple soil pits, choosing sampling intervals within individual pits,

etc.) with appropriate averaging relative to scales of disturbances and mixing remains an important open question.

Momentarily assuming that mixing conditions are reasonably reflected by the expected particle OSL age profile $\hat{m}_1(\hat{z})$, then the results above bear on the practical question of variability in these expected ages as a consequence of small sample sizes. As a point of reference, Heimsath et al. (2002) sampled an average of 41 quartz grains from each of one to three vertical positions within four soil pits. Of the total 10 samples, on average 19 grains had finite OSL ages. Johnson et al. (2014) analyzed 42–49

grains from each of five intervals in a single soil pit. Considering only grains with finite OSL ages, the sample size $N_s$ from each vertical interval is about $20 - 50$ in these examples. Regardless of the form of the distribution of finite particle OSL ages with variance $\sigma^2$ within each interval (Figure 8), the central limit theorem suggests that the standard error $s_e$ of the estimate of the mean is $s_e \approx \sigma/\sqrt{N_s}$, or, in dimensionless form, $\hat{s}_e \approx \hat{\sigma}/\sqrt{N_s}$.

Let is assume that within a small interval of $\hat{z}$, $\hat{\sigma}^2 = \hat{m}_2(\hat{z})$ from Eq. (45) or Eq. (46). We may then write

$$\hat{s}_e(\hat{z}) \approx \pm\sqrt{\frac{\hat{m}_2(\hat{z})}{N_s}}. \tag{52}$$

This yields an estimate of $\hat{s}_e(\hat{z})$ depending on the intensity and structure of mixing in relation to the sample size $N_s$, and represents uncertainty in the mean value $\hat{m}_1(\hat{z})$ that is in addition to analytical uncertainty associated with single-grain OSL age estimates. The standard errors $\hat{s}_e(\hat{z})$ for uniform and nonuniform mixing are similar, although nonuniform mixing generally yields smaller values of $\hat{s}_e(\hat{z})$. The well-known formula Eq. (52) suggests that, in order to obtain a standard error $\hat{s}_e(\hat{z})$ of

specified magnitude within a small interval at position $\hat{z}$ requires that $N_s \approx \hat{m}_2(\hat{z})/s_e^2$. Because $\hat{m}_2(\hat{z})$ increases with depth (Figure 9), uncertainty in the estimate of the expected value $\hat{m}_1(\hat{z})$ increases with depth for a given sample size $N_s$, as directly reflected in the data of Heimsath et al. (2002) and Johnson et al. (2014). Stated another way, there may be value in judiciously varying $N_s$ with depth when faced with a research budget that limits the total number of single-grain OSL age analyses. We note, however, that this uncertainty associated with sample size cannot be distinguished from effects of any legacy of

disturbances as described above.

The results of the numerical simulations as depicted in Figures 3, 4, 6 and 7 provide an important perspective on the nature of production of $^{10}$Be and OSL age in relation to particle transport and mixing, and the associated structuring of the profiles $n(z)$ and $m_1(z)$. We note that the points in Figures 3 and 4 represent large samples drawn from the joint probability density $f_{n_p,z}(n_p, z, t)$, and the points in Figures 6 and 7 represent samples drawn from the joint probability density $f_{A_p,z}(A_p, z, t)$.

With respect to $f_{n_p,z}(n_p, z, t)$, at any instant a particle within the $n_p - z$ domain only can move in the positive $n_p$ direction due to its accumulation of $^{10}$Be atoms (neglecting decay). Similarly, with respect to $f_{A_p,z}(A_p, z, t)$, a particle within the $A_p - z$ domain only can move in the positive $A_p$ direction due to its accumulation of OSL age. This means that the distribution $f_{n_p}(n_p, z)$ or $f_{A_p}(A_p, z)$ at any position $z$ as depicted in Figures 5 and 8 is at all instants being uniformly advected in the

positive $n_p$ or $A_p$ direction. In both cases, particles at any instant may move in either the positive or negative $z$ direction due to their random-walk motions.

Combining Eq. (4) and Eq. (14), neglecting particle volume and the decay of $^{10}$Be, and assuming steady conditions,

$$-\frac{\partial}{\partial z}(q_A + q_D) - P(z)\frac{\partial f_{n_p,z}(n_p, z)}{\partial n_p} = 0 \,, \tag{53}$$

where $q_A$ and $q_D$ denote the advective and diffusive parts of the flux. Similarly, combining Eq. (20) and Eq. (23),

$$-\frac{\partial}{\partial z}(q_A + q_D) - S\frac{\partial f_{A_p,z}(A_p, z)}{\partial A_p} = 0 \,. \tag{54}$$

These highlight how production at any position within the $n_p - z$ or $A_p - z$ domain is exactly balanced by the local, combined effects of particle advection and diffusion. Consider the density $f_{n_p,z}(n_p, z)$. With reference to Figure 5, at all locations $(n_p, z)$ where the derivative $\partial f_{n_p,z}(n_p, z)/\partial n_p < 0$, the effect of production is to increase the $^{10}$Be content at these loca-
tions in proportion to the production rate $P(z)$ and the magnitude of this derivative; and at locations where the derivative $\partial f_{n_p,z}(n_p, z)/\partial n_p > 0$ the effect of production is to decrease the $^{10}$Be content at these locations. The variation in $q_A$ and $q_D$ with respect to $z$ must be such that their combined divergence balances these effects of production. In turn, consider the density $f_{A_p,z}(A_p, z)$. With reference to Figure 8, at all locations $(A_p, z)$ where the derivative $\partial f_{A_p,z}(A_p, z)/\partial A_p < 0$ the effect of particle aging is to increase the OSL age content at these locations in proportion to the magnitude of this derivative; and at
locations where the derivative $\partial f_{A_p,z}(A_p, z)/\partial A_p > 0$ the effect of particle aging is to decrease the OSL age content at these locations. Variations in $q_A$ and $q_D$ with respect to $z$ must then compensate these effects.

We normally envision that local production of a quantity implies a local increase in the quantity. But this is not necessarily so when viewed in the $n_p - z$ or $A_p - z$ domain. Only when the production is averaged via integration over the $n_p$ or $A_p$ domain, as in Sections 3.2.2 and 3.3.2, does a production term emerge as normally envisioned. This point further highlights a key
idea underlying the formulation, that extensive and intensive particle properties are not in themselves subject to advection and diffusion, but rather, are merely carried with the particles as these undergo advection and diffusion with respect to $z$. Indeed, the production terms in Eq. (53) and Eq. (54) represent only advection over the $n_p$ and $A_p$ domains, not diffusion (mixing) over these domains.

The numerical simulations suggest that the overall particle OSL age distribution is approximately exponential (Figure 10),
consistent with field data (see data of Heimsath et al. (2002) as described by Furbish et al. (2018b)). This result awaits a theoretical explanation. Meanwhile, as described by Furbish et al. (2018b), the distribution $f_{T_r}(T_r)$ of the return times $T_r$ between successive encounters of a particle with the soil surface is expected to be a power-law distribution with an undefined mean (Redner, 2001) for the idealized situation involving uniform Gaussian mixing in a vertically unbounded domain, in the absence of upward advection. Because the OSL age of a particle increases at the same rate as its (eventual) return time, the
distribution of OSL ages also is likely to be a power-law distribution in this situation. However, upward advection (with surface erosion) combined with a finite soil thickness have the effect of strongly tempering this distribution, yielding an approximate exponential form. Further tempering is provided with nonuniform mixing, where diffusion decreases with depth then vanishes at the soil-saprolite interface. This behavior of particle OSL ages is entirely analogous to the exponential tempering of the

power-law distribution of residence times of particles undergoing burial and exhumation in a stream channel, where a finite sediment thickness limits the depth of burial. At long times the particles fully explore the accessible thickness, and a finite (unchanging) average residence time emerges (Voepel et al., 2013).

The emergence of a maximum average OSL age $\hat{M}_1$ at an intermediate Péclet number $Pe \sim 1$ (Figure 11) is in direct contrast with the two-dimensional case involving downslope transport by creep without surface erosion (Furbish et al., 2018b), where the average OSL age monotonically decreases with decreasing $Pe$. In this case, at large $Pe$, OSL particles remain near the surface (as in the one-dimensional case), but they can accumulate large ages before exiting the soil mantle downslope. Moreover, in the one-dimensional case, that the average OSL age is a fraction of the mean particle residence time lends support to the idea of defining two distinct populations of OSL tracers (Heimsath et al., 2002; Furbish et al., 2018b), those with finite age and those that are saturated, having an "infinite" age, inasmuch as the mean residence time is much smaller than the determinable OSL age limit (Murray and Olley, 2002).

That the numerical simulations mimic analytical solutions for the benchmark situation of a one-dimensional mean motion involving both uniform and nonuniform mixing with varying mixing intensities lends confidence in applying the numerics to more complicated situations. Such situations might be motivated by questions concerning consequences of transient conditions of surface erosion and soil production, aeolian inputs to the soil, particle weathering in relation to particle aging, accumulation of luminescence signals with nonuniform dose rates, and the structuring of tracer particles under depositional conditions. Our experience suggests the need to implement the numerics of boundary conditions carefully, ensuring consistency with global particle conservation.

Here we return to our starting point. Our use of the Fokker-Planck equation assumes Gaussian diffusion of tracer particles. As described above, this is a parsimonious choice whose consequences, and veracity, must be judged by its consistency with measurable outcomes of mixing, including profiles of CRN concentrations and OSL ages as emphasized here, but possibly to include other soil properties. We suggest that a Gaussian model of particle mixing is robust inasmuch as this mixing behavior is insensitive to the form of the probability distribution of particle displacements, $f_r(r)$, so long as this distribution is not heavy-tailed. We further emphasize that the effective particle diffusivity may actually represent motions involving a mixture of characteristic length scales and associated frequencies of occurrence in settings involving both biotic and abiotic disturbances. We also acknowledge that it may be more appropriate to consider some disturbances, for example, macro-disturbances by tree throw and fossorial animals, as having the effect of stirring rather than mixing, where homogenization occurs at length scales comparable to the mechanically active soil thickness (see next section). This points to the need for a clearer understanding of the spatiotemporal structure of mixing motions in adopting more sophisticated (i.e., non-Gaussian) models of mixing behavior. The goal is to understand the information content of tracers aimed at constraining mechanical formulations of transport and mixing, notably in relation to soil creep. The one-dimensional benchmark situation described here is a key starting point due to the lessons it offers.

## 6.2 Assessing the intensity and depth dependence of mixing

Here we focus on results for the one-dimensional benchmark case (Section 4, Figure 2) — specifically the profiles of expected [10]Be concentrations and particle OSL ages — to suggest constraints on assessing the intensity and depth dependence of mixing. For ease of comparison, we collect these profiles from Figures 3 and 4, and from Figures 6 and 7, and combine them in Figures 12 and 13.

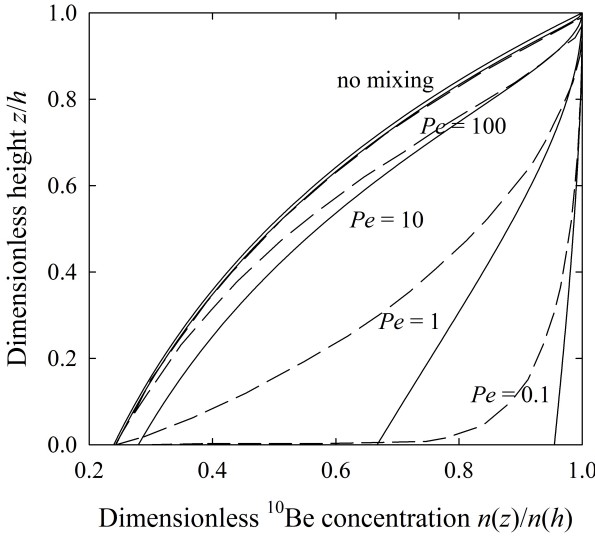

**Figure 12.** Plot of dimensionless expected concentration $\hat{n}(\hat{z}) = n(z)/n(h)$ of [10]Be atoms versus dimensionless height $\hat{z} = z/h$ with uniform mixing (solid lines) and nonuniform mixing (dashed lines) as these vary with the Péclet number $Pe = Wh/K_z$.

As described above, these profiles systematically vary with the Péclet number, $Pe = Wh/K_z$. In the case of [10]Be concentrations, the profile converges to the exponential solution provided by Lal (1991) for weak mixing (large $Pe$), and it converges to a uniform value equal to the surface concentration for strong mixing (small $Pe$). In the case of expected particle OSL ages, the profiles vary approximately linearly with depth, and converge to a uniform value close to zero for strong mixing.

10 Not surprisingly, with weak mixing the [10]Be and OSL profiles for uniform and nonuniform mixing are virtually indistinguishable (Figures 12 and 13), as the profiles in this case are mostly determined by the mean motion. Similarly, with strong mixing the [10]Be profiles are not markedly different except near the base of the soil column, and the OSL age profiles are nearly the same. Significant differences in the profiles appear only in the presence of intermediate mixing intensities. The essence of these differences at intermediate intensities ($Pe \sim 1$) arises from how rapidly the particle diffusivity decreases with increasing

15 depth (Sections 5.1 and 5.2). Thus, the forms of the profiles might change in detail in the presence of a more complicated (e.g., nonlinear) mixing structure. Nonetheless, these results suggest that [10]Be and OSL age profiles may help constrain the mixing structure in the presence of intermediate mixing intensities, albeit depending on the resolution of measurements.

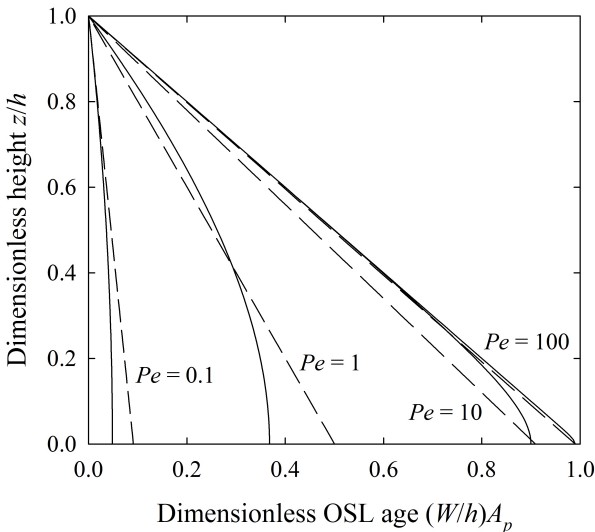

**Figure 13.** Plot of dimensionless expected particle OSL age $\hat{A}_p(\hat{z}) = (W/h)A_p(z)$ versus dimensionless height $\hat{z} = z/h$ with uniform mixing (solid lines) and nonuniform mixing (dashed lines) as these vary with the Péclet number $Pe = Wh/K_z$.

These profiles highlight that uniform particle mixing is not synonymous with the idea of complete mixing, and why a uniform profile of [10]Be concentration or particle OSL age does not necessarily indicate the presence of uniform mixing. Whereas "uniform mixing" refers to the mixing structure wherein the statistical qualities of particle random walks are independent of vertical position, "complete mixing" refers to an idea from reservoir theory (Bolin and Rodhe, 1973), that particle mixing within a specified control volume is sufficiently thorough that the probability of a particle exiting the volume is independent of its residence time in the volume (Bolin and Rodhe, 1973; Furbish et al., 2018a) — an idea that is strongly conditioned by the geometry of particle motions, specifically, the proximity of the inflow and outflow locations relative to the particle trajectories, and the degree of mixing between these locations (Bolin and Rodhe, 1973). Both uniform and nonuniform mixing yield uniform [10]Be and OSL profiles in the limit of $Pe \to 0$. That said, complete particle mixing within soils is mechanically unlikely, a point that is consistent with available [10]Be and OSL data concerning creeping soils (Furbish et al., 2018a, 2018b), and deserving reexamination in interpreting [10]Be profiles with respect to surface ages and denudation rates (Schaller et al., 2009). This point also is consistent with the idea of depth dependent mixing (Humphreys and Field, 1998; Cousins et al., 1999; Roering, 2004; Wilkinson and and Humphreys, 2005; Wilkinson et al., 2009; Johnson et al., 2014; Gray, 2018), in which the local intensity of mixing declines with depth.

Here we step back and look at published data. We first note that, whereas our benchmark case involves a steady one-dimensional mean motion, available field-based measurements of [10]Be concentrations and OSL particle ages mostly pertain to transient conditions or involve two-dimensional downslope soil transport. One cannot make a direct comparison between tracer profiles sampled on sloping surfaces and the one-dimensional results depicted in Figures 12 and 13. For example, the upper boundary conditions examined here are quite different from those in the two-dimensional case. One effect of these differences

is directly reflected by the column-averaged OSL age as this varies non-monotonically with the Péclet number $Pe$ (Figures 10 and 11) versus the monotonic variation of this quantity with $Pe$ for two-dimensional particle motions (Furbish et al., 2018b, Figure 6 therein). Nonetheless, in comparing our results with those presented in Furbish et al. (2018b, Figures 4 and 5 therein), it is clear that the basic forms of profiles resulting from one-dimensional and two-dimensional transport systematically vary in

like manner with the intensity of mixing, as characterized by the Péclet number $Pe$.

As an important backdrop to the benchmark case examined here, $^{10}$Be profiles from a flight of five marine terraces near Santa Cruz, California, illustrate the continued accumulation of $^{10}$Be atoms with increasing terrace age within the mixed soil and underlying undisturbed material, under the condition of negligible surface erosion (Perg et al., 2001, Figures 2 and 4 therein; Granger and Riebe, 2014, Figure 9 therein). Near-surface concentrations are relatively uniform, and in three cases (terraces 1, 3

and 5) decline toward the value at the base of the assumed mixing depth, suggesting $Pe \sim 1$ and likely an associated decline in mixing intensity. Concentrations are mostly centered about a vertically averaged value that is less than the surface concentration that would occur with steady surface erosion. Similarly, as noted by Furbish et al. (2018b), uniform concentrations of $^{10}$Be in weakly developed soils on the crests of moraines near Pinedale, Wyoming, suggest well mixed conditions near the surface (Schaller et al., 2009), although there is inconsistency with expected concentrations based on the formulation of Lal and Chen

(2005) for the well mixed case; there also is uncertainty in the calculated lowering rates and mixing depths, and the sites may represent transient conditions.

Relatively uniform $^{10}$Be profiles from hillslopes in the Great Smokey Mountains reflect strongly mixed conditions at the sample locations (Jungers et al., 2009, Figure 7 therein; reproduced in Anderson, 2015, Figure 14 therein), likely due to effects of tree throw and other bioturbation events that stir the soil over much of its $\sim$60 cm thickness. Within the context of the

analysis above, these conditions suggest that $Pe < 1$. Similarly, five profiles sampled on hillslopes at Gordon Gulch, Colorado, display a mixture of conditions, varying from relatively uniform concentrations ($Pe < 1$) to an approximately linear variation with depth ($Pe > 1$) (Foster et al., 2015, Figure 7 therein; reproduced in Anderson, 2015, Figure 14 therein). In turn, three profiles measured along a 100 m catena flow line in a soil developed from granitic bedrock on Osborn Mountain, Wyoming (Small et al., 1999, Figure 6 therein; reproduced in Anderson, 2015, Figure 14 therein), reflect conditions consistent with

$Pe \gtrsim 1$ (Furbish et al., 2018b), where relatively uniform concentrations in the upper parts of the profiles then decrease in the lower one third.

An OSL age profile $A_p(z)$ based on single quartz grains collected from a bioturbated soil developed on a basalt flow on the Denna Plain in northeast Queensland, Australia (Johnson et al., 2014, Figure 2 therein; Furbish et al., 2018b, Figure 10 therein), most closely matches the benchmark case described here. This profile suggests an approximately linear increase in

OSL ages with depth, as in Figure 13. In addition, the sampled quartz grains likely were added to the soil at its surface, a boundary condition that is consistent with the theoretical formulation (Furbish et al., 2018b; Appendixes C and D). Moreover, the OSL ages are only a fraction of the estimated mean residence time at this site, consistent with moderate to strong mixing (Furbish et al., 2018b). Although mixing at this site likely varies with depth, the similarity between profiles in Figure 13 suggests that the mixing structure cannot be distinguished. Similarly, the OSL age profiles $A_p(z)$ reported by Heimsath et al.

(2002, Figure 1 therein; Furbish et al., 2018b, Figure 8 therein) based on single grains of quartz collected from a hillslope

with nonuniform soil thickness over granitic bedrock in the Nunnock River catchment, Australia, suggest an approximately linear increase in OSL ages with depth. Although involving downslope transport, the profiles are consistent with strong mixing (small $Pe$), possibly including macro-disturbances (Heimsath et al., 2002). Moreover, the distribution of all particle OSL ages is approximately exponential with an average age that is much smaller than the calculated mean soil residence time, consistent

with strong mixing (Furbish et al., 2018b; Figure 10).

In all cases summarized above, the profiles suggest moderate ($Pe \gtrsim 1$) to strong ($Pe < 1$) mixing. Distinguishing between uniform and non-uniform mixing likely will require higher resolution sampling than reported in these cases. Our own bias is that many if not most settings with significant mixing by bioturbation or the effects of freezing and thawing likely involve depth dependent mixing. At least for the one-dimensional case examined here, CRN profiles are capable of revealing mixing

intensity and possibly mixing structure for $Pe \sim 1$. In contrast, OSL profiles are capable of revealing mixing intensity, but not likely mixing structure.

To our knowledge there are no available measurements of profiles of $^{10}$Be concentrations and OSL ages taken together. We suggest that there is merit in doing just this as a means to provide a more demanding test of formulations of transport and mixing (Furbish, 2003; Roering et al., 2004). We also reiterate that our results provide an analytical benchmark for assessing

the veracity of emerging numerical methods aimed at simulating particle transport and mixing, to include Eulerian-Lagrangian descriptions of particle motions that might incorporate individual detrital grain CRN concentrations (Codilean et al., 2010) as well as fully treating the effects of a nonuniform radiation dose field. This includes simulations that start from probabilistic, physically based formulations of total luminescence intensities as measured by portable OSL readers (Gray et al., 2017; Gray, 2018), as an addition to multi-grain and single-grain analyses aimed at extracting particle burial ages. Such measurements

may be capable of revealing mixing structure, as well as intensity, from relatively high resolution sampling since particles involved in accumulating luminescence signals are likely to be more uniformly distributed within the soil column relative to those possessing finite OSL ages (Appendix C).

*Code availability.* The code for simulating particle motions is written for Matlab, and is available by request from any of the authors.

## Appendix A: Rarefied versus continuum conditions

To further illustrate the significance of the probabilistic formulation of conservation in relation to rarefied versus continuum conditions, here we start with the familiar example of Brownian motion, the initial formal description of which is separately attributable to Einstein (1905) and von Smoluchowski (1906). With reference to Figure A1, let $x$ denote a coordinate along which Brownian particles take one-dimensional random walks, where $x$ extends indefinitely in the positive and negative directions about the origin $x = 0$. Suppose that a particle starts at the origin at time $t = 0$, and with equal probability moves

in the positive or negative direction during successive small intervals $\mathrm{d}t$. By the definition of a random walk, the motion of the particle — specifically its expected position $x$ after an interval of time $t > 0$ — can be predicted only in a probabilistic

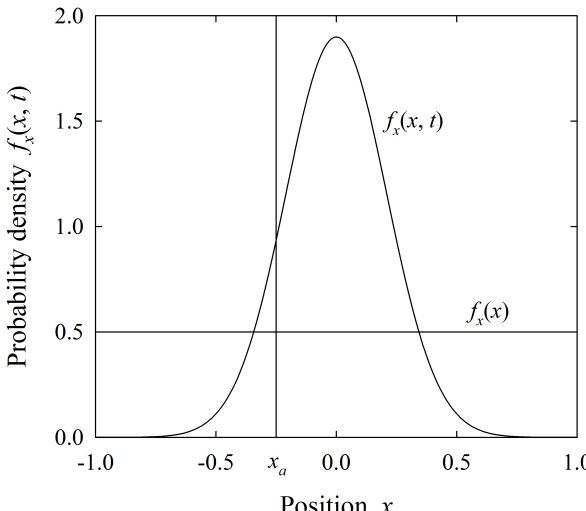

**Figure A1.** Plot of coordinate position $x$ of particle undergoing a random walk motion showing: Gaussian distribution $f_x(x,t)$ of expected positions at time $t$ as the solution, Eq. (A2), of the Fokker-Planck equation, and the actual (example) particle position $x = x_a$; and uniform steady-state distribution $f_x(x) = 1/2$ for a bounded domain such that $-1 < x < 1$.

sense. Namely, letting $f_x(x,t)$ denote the probability density function of possible positions $x$, then this density satisfies a Fokker-Planck equation involving only its diffusion term:

$$\frac{\partial f_x(x,t)}{\partial t} = \kappa_x \frac{\partial^2 f_x(x,t)}{\partial x^2}, \tag{A1}$$

where $\kappa_x$ denotes the particle diffusivity. The solution of Eq. (A1) is the Gaussian distribution with mean $\mu_x = 0$, namely,

$$5 \quad f_x(x,t) = \frac{1}{\sqrt{4\pi\kappa_x t}} e^{-x^2/4\kappa_x t}. \tag{A2}$$

For this highly rarefied system involving a single particle, we can only offer probabilistic predictions of its position at time $t$. For example, we may confidently state that with probability $p = 1/2$ the particle is either at a position $x < 0$ or at a position $x > 0$. Or, we may state that with probability $p \approx 0.68$ the particle is within the domain defined by plus one and minus one standard deviations about the mean position, namely, $-\sqrt{2\kappa_x t} < x < +\sqrt{2\kappa_x t}$. For this single-particle system (realization),

10    the actual particle position $x_a$ is represented by a Dirac distribution $\delta(x_a - x, t)$ (Figure A1), but this cannot be predicted deterministically.

     Let us now imagine an arbitrarily great number $N$ of identical, independent particles that start at the origin $x = 0$ at time $t = 0$, each undergoing a random walk during $t > 0$. When viewed together, the distribution of these particles at time $t = 0$ is given by the Dirac distribution, namely, $f_x(x,0) = \delta(x)$. At any time $t > 0$ these particles are distributed according to Eq. (A2).

15    That is, because $N$ is arbitrarily large, the proportion of particles within any small interval $x$ to $x + \mathrm{d}x$ closely matches what is predicted by Eq. (A2), namely $f_x(x,t)\mathrm{d}x$, such that in the limit of $\mathrm{d}x \to 0$ the actual distribution of positions $x$ varies smoothly

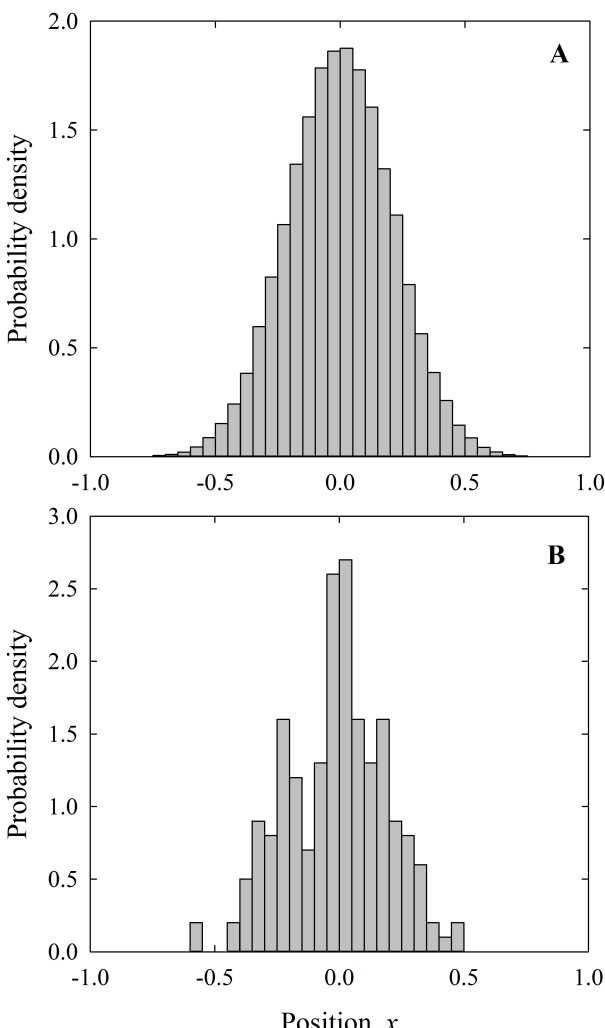

**Figure A2.** Histograms of particle positions $x$ at time $t$ for one system showing that: (A) with a great number $N$ of particles representing a continuum condition this histogram converges to the smooth Gaussian distribution in Figure A1 as $dx \to 0$; in this example $N = 100{,}000$; and (B) with a modest number of particles representing a rarefied condition this histogram is irregular and discontinuous; in this example $N = 200$.

(continuously) and converges to Eq. (A2) (Figure A2). In contrast to the highly rarefied single-particle system in the previous example, we may thus assume that this great number of particles, occurring in one system (realization), satisfies the continuum hypothesis. Nonetheless, upon randomly selecting a single particle from this system, we still can only offer probabilistic predictions of its position at time $t$ — as in the example above involving a system with a single particle. Moreover, note that the continuous distribution of positions $x$ realized at time $t$ for this one system involving a great number $N$ of particles is

identical to the distribution that would be realized upon pooling the $x$ positions at time $t$ associated with a great number $N$ of independent systems, each involving a single particle.

Now select a system with a modest number $N$ of particles such that conditions are rarefied. By this we mean that, after some time $t$, the actual distribution of particle positions $x$ is at best represented by an irregular histogram that roughly appears Gaussian, but is decidedly discontinuous (Figure A2). Moreover, any realization involving $N$ particles possesses a similar irregular form at time $t$, and no two are the same. In effect, each realization represents a sample of size $N$ drawn from an imagined population represented by Eq. (A2). Also note that each realization involving $N$ particles at time $t$ is the same as $N$ realizations, each involving a single particle, when viewed collectively at time $t$.

Let us now consider a great number $N_e$ of independent but nominally identical systems — an ensemble — at any fixed time $t$, where each system contains $N$ particles, large or small. We now wish to describe the ensemble expected conditions. To envision this, consider any small interval $x$ to $x + \mathrm{d}x$. If $N = 1$ as in the first example above, then $f_x(x,t)\mathrm{d}x$ is just the proportion of the $N_e$ systems containing a particle within $x$ to $x + \mathrm{d}x$ at time $t$. Note that this is identical to the result above involving an individual system containing a great number $N = N_e$ of particles. If instead each system involves a great number $N$ of particles, then $f_x(x,t)\mathrm{d}x$ simply becomes the expected proportion of the $N$ particles within $x$ to $x + \mathrm{d}x$ at time $t$, where the expectation is calculated over the $N_e$ systems. And note that this outcome is identical to the proportion of $N \times N_e$ independent systems, each involving a single particle, which contain a particle within $x$ to $x + \mathrm{d}x$ at time $t$. In either case, the expected proportion within the interval is the same. Moreover, we reach the same conclusion in considering a great number $N_e$ of systems, each involving a modest number $N$ of particles. Thus, when calculated over a great number of systems for all intervals $\mathrm{d}x$, then in the limit of $\mathrm{d}x \to 0$, the continuous function, Eq. (A2), is retrieved. The key points are these: First, whether $N$ is relatively small (representing a rarefied condition) or $N$ is large (representing a continuum condition), the ensemble expected behavior represented by Eq. (A2) applies equally to both conditions in a probabilistic sense. Second, if $N$ is small, then Eq. (A2) represents the ensemble expected behavior, not the actual behavior of any one system (realization); and if $N$ is large, then the actual behavior of the system is expected to converge to the smooth ensemble behavior represented by Eq. (A2).

To complete the picture, suppose that the $x$ domain in Figure A1 is bounded such that $-1 < x < 1$. Particles that reach these boundaries are "reflected" and remain within the domain, continuing their random walks. In the limit of $t \to \infty$, the probability density of particle positions $x$ reaches a steady-state form, that is, $\partial f_x(x,t)/\partial t \to 0$ such that $f_x(x,t) \to f_x(x)$. In this limit, Eq. (A1) becomes $\mathrm{d}^2 f_x(x)/\mathrm{d}x^2 = 0$. Moreover, the probability flux $q_x = -\kappa_x \mathrm{d}f_x(x)/\mathrm{d}x = 0$ at all positions $x$, which means that $\mathrm{d}f_x(x)/\mathrm{d}x = 0$. These constraints together with the fact that the distribution $f_x(x)$ must integrate to unity yield the result that $f_x(x) = 1/2$ over the bounded domain (Figure A1). That is, the expected distribution $f_x(x)$ is uniform. As with the unsteady problem described above, a modest number $N$ of particles representing rarefied conditions in any one realization is at best represented by an irregular histogram that roughly appears uniform, but is decidedly discontinuous (Figure A3). Moreover, at an arbitrary later time, the resulting distribution (histogram) would be just as irregular; it does not become smoother with increasing time. As above, the expected continuous steady-state distribution is retrieved when expected values are calculated over a great number $N_e$ of systems.

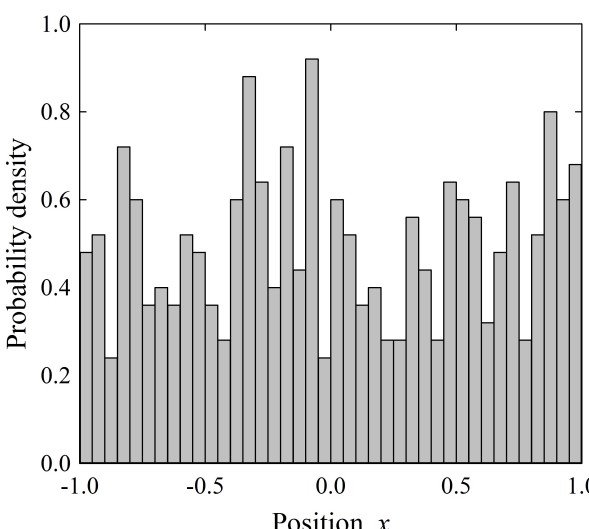

**Figure A3.** Histogram of particle positions $x$ at time $t \to \infty$ for one system showing that with a modest number of particles representing a rarefied condition this histogram is irregular and discontinuous; in this example $N = 500$.

To place these ideas within the context of soil tracer particles, including the practical assessment of rarefied versus continuum conditions, let us now imagine a soil column of thickness $h = 1$ m. Suppose that we wish to be able to describe the probability density $f_z(z)$ of tracer particle positions $z$ as a smooth function at a resolution of, say, 1 cm. That is, we are aimed at a function that "looks" smooth in proceeding along $z$ from each 1-cm increment to the next. Choosing a finer resolution would not make
sense because then we are approaching the scale of individual particles. On the other hand, choosing a significantly larger resolution (e.g., 10 cm) would represent a loss of information at scales of possible interest.

Let us now measure the number of tracer particles within each of 100 cubic centimeters representing a column of soil, that is, a column with horizontal dimensions $X = Y = 1$ cm. For illustration, let us assume that the expected number of particles within each cubic centimeter of soil is 10. The expected number of particles within the entire column therefore is $N = 1,000$.
In this example the expected proportion $f_z(z)\mathrm{d}z$ with $\mathrm{d}z = 1$ cm is 0.01.

Now imagine that, due to randomness in the particle mixing process, the number of tracer particles $n(z)$ varies from one increment $\mathrm{d}z = 1$ cm to the next about the expected average of 10. Assuming for illustration that this variability is spatially random at the centimeter scale, we may formally draw values of $n(z)$ from a binomial distribution (or approximate this using a Poisson distribution or a normal distribution) with a mean of $\mu = 10$ to populate each increment in the soil column. The total
expected number of particles remains $N = 1,000$, so we may calculate the associated proportions $n(z)/N$ to represent the function $\hat{f}_z(z)$ (Figure A4). Notice that $\hat{f}_z(z)$ fluctuates significantly about the expected value $f_z(z)\mathrm{d}z = 0.01$.

Consider first the idea of describing this one realization as a continuum, that is, where we might imagine that $\hat{f}_z(z)\mathrm{d}z$ is smooth. To do this, we apply a boxcar moving average of width $L$ over the spatial series in Figure A4. We may think of this width $L$ as the averaging length defining a continuum physical point, as described in the text. Notice that in this example, $L$

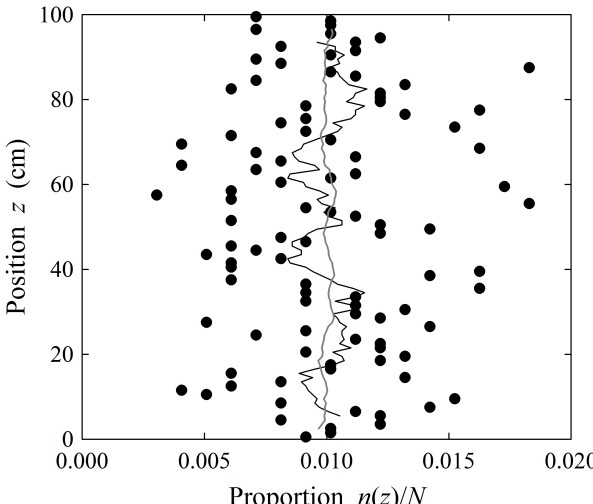

**Figure A4.** Plot of proportion $n(z)/N = \hat{f}_z(z)$ versus position $z$ for the example of $N = 1{,}000$ particles and $X = Y = 1$ cm with expected proportions of $f_z(z)\mathrm{d}z = 0.01$ (black dots), then smoothed with boxcar filter with $L = 10$ cm (black line); and proportion for $N = 100{,}000$ with $X = Y = 10$ cm and $L = 4$ cm (gray line).

must approach a significant proportion of the soil thickness $h$ before a relatively smooth function emerges — that is, a function that we would not be uncomfortable in taking its derivative to define a gradient having physical significance with respect to particle transport. However, there may be uncertainty in the fidelity of this operation. For example, in the situation where $n(z)$ is not uniform, then because averaging of this sort represents a low-pass filter, averaging may obscure physically meaningful variations of interest.

Consider now the idea of expanding either (or both) the horizontal dimensions $X$ and $Y$ of the soil column. In this situation both $n(z)$ and $N$ increase. Fluctuations $\hat{f}_z(z)$ about the expected value $f_z(z)$ systematically decrease with increasing $N$. Using a boxcar moving average, smoothness in $\hat{f}_z(z)$ emerges with a smaller length $L$ (Figure A4) relative to the previous situation involving smaller $XY$ and $N$. That is, $\hat{f}_z(z)$ looks more like a continuous function at finer resolution. This is the reason for

stating in the text that one hopes to sample over a sufficiently large area $XY$ to obtain an approximately smooth distribution in any one realization. Note, however, that this smoothness is scale dependent, as it requires a sufficiently large area $XY$, and assumes that the process of mixing is uniform over $x$ and $y$. But this also means that variations with respect to $x$ or $y$ in the full three-dimensional probability density $f_{x,y,z}(x,y,z)$ of particle positions $x$, $y$ and $z$ cannot be resolved in the situation where the mixing process varies with $x$ or $y$ (e.g., two-dimensional transport and mixing).

Finally, consider the idea of taking an ensemble average, where we return to the small 1 cubic centimeter sampling volumes as described above. If at any geometrically similar position $(x,y,z)$ within a great number of nominally identical but independent systems the number of particles within the volume $\mathrm{d}x\mathrm{d}y\mathrm{d}z$ varies from one system (realization) to the next, the central limit theorem guarantees that the ensemble average of this number converges to its ensemble expected value. It immediately

follows that the ensemble expected distribution $f_{x,y,z}(x,y,z)$ is a smooth continuous function; or in one dimension, $f_z(z)$ for specified $XY$ is a smooth continuous function.

With respect to developments in the text, the Fokker-Planck equation describes the time evolution of the probability density $f_z(z,t)$. The formulation does not assume either rarefied or continuum conditions. It is indifferent to these conditions, yet equally applicable to both. As described in the text, the Fokker-Planck equation is a special case of the Master equation, a general statement of conservation of probability that is independent of scale, and which is the basis of more familiar statements of conservation when probability is reinterpreted in terms of, for example, mass, momentum or energy. If in an individual system (realization) the continuum hypothesis is satisfied (a condition that is independent of the probabilistic basis of the Master equation or the Fokker-Planck equation), then the probabilistic formulation based on ensemble expected conditions and its continuum counterpart are essentially one and the same. If, however, the continuum hypothesis is not satisfied, then one cannot defensibly start with a continuum equation, but instead must appeal to a probabilistic formulation of ensemble expected conditions (in order to justify the use of continuously differentiable equations), with the proviso that any prediction of the behavior of an individual (rarefied) system is probabilistic in nature.

## Appendix B: Conservation of expected number concentration of $^{10}$Be atoms

Assuming $^{10}$Be production is due to spallation (Gosse and Phillips, 2001), the number concentration $n(z,t)$ of $^{10}$Be atoms satisfies a Foker-Planck-like equation, namely,

$$\frac{\partial n(z,t)}{\partial t} = -\frac{\partial}{\partial z}\left[w(z,t)n(z,t) - \kappa_z(z,t)\frac{\partial n(z,t)}{\partial z}\right]$$

$$+P_0 e^{-(h-z)/l_s} - \lambda n(z,t), \tag{B1}$$

where $w(z,t)$ denotes the ensemble averaged particle velocity and $\kappa_z(z,t)$ denotes the ensemble expected particle diffusivity (Furbish et al., 2009b, 2018a, 2018b). For steady conditions involving a one-dimensional mean motion, $\partial n(z,t)/\partial t = 0$, $n(z,t) \rightarrow n(z)$ and $w(z,t) \rightarrow W$. Assuming that the half-life of $^{10}$Be is much greater than the mean residence time, $h/W$, of target quartz particles, then Eq. (B1) becomes

$$\frac{\mathrm{d}}{\mathrm{d}z}\left[Wn(z) - \kappa_z(z)\frac{\mathrm{d}n(z)}{\mathrm{d}z}\right] = P_0 e^{-(h-z)/l_s}. \tag{B2}$$

We first rewrite Eq. (B2) in terms of the flux $q_z(z) = Wn(z) - \mathrm{d}n(z)/\mathrm{d}z$, namely,

$$\frac{\mathrm{d}q(z)}{\mathrm{d}z} = P_0 e^{-h/l_s} e^{z/l_s}. \tag{B3}$$

Vertically integrating Eq. (B3) over the soil thickness,

$$\int_0^h \frac{\mathrm{d}q(z)}{\mathrm{d}z}\,\mathrm{d}z = P_0 e^{-h/l_s}\int_0^h e^{z/l_s}\,\mathrm{d}z. \tag{B4}$$

With fixed boundaries, this yields

$$q_z(h) = q_z(0) + P_0 l_s \left(1 - e^{-h/l_s}\right). \tag{B5}$$

This says that the flux $q_z(h)$ of $^{10}$Be atoms across the soil surface and removed by erosion is equal to the rate at which atoms enter the soil column at its base, $q_z(0) = W n(0)$, plus the total rate at which they are produced within the column. In this steady problem, no information is available regarding the vertically averaged concentration.

In this problem, the concentration $n(0)$ at the soil-saprolite interface is obtained by solving the purely adective form of Eq. (B2) and using the boundary condition that $n(-\infty) = 0$. The result is

$$n(0) = \frac{P_0 l_s}{W} e^{-h/l_s}. \tag{B6}$$

In turn, the flux $q_z(0)$ is

$$q_z(0) = P_0 l_s e^{-h/l_s}, \tag{B7}$$

and the flux $q_z(h)$ is

$$q_z(h) = P_0 l_s. \tag{B8}$$

As described below, the concentration $n(h)$ at the soil surface depends on what is assumed about the contributions to the flux $q_z(h)$ at this surface. We now consider how the concentration profile $n(z)$ differs with uniform versus nonuniform mixing.

## B1   Uniform mixing

With uniform mixing ($\kappa_z = K_z$), we start by integrating Eq. (B3) to give

$$q_z(z) = P_0 l_s e^{-h/l_s} e^{z/l_s} + C_1. \tag{B9}$$

The lower boundary condition, Eq. (B7), then gives $C_1 = 0$. Using this result together with $q_z(z) = W n(z) - K_z \mathrm{d}n(z)/\mathrm{d}z$ then leads to

$$\frac{\mathrm{d}n(z)}{\mathrm{d}z} - \frac{W}{K_z} n(z) = -\frac{P_0 l_s}{K_z} e^{-h/l_s} e^{z/l_s}. \tag{B10}$$

We now simplify the notation and rewrite Eq. (B10) as

$$\frac{\mathrm{d}n(z)}{\mathrm{d}z} - A n(z) = -B e^{z/l_s}, \tag{B11}$$

with

$$A = \frac{W}{K_z} \quad \text{and} \quad B = \frac{P_0 l_s}{K_z} e^{-h/l_s}. \tag{B12}$$

A general solution of Eq. (B11) is

$$n(z) = \frac{B l_s}{A l_s - 1} e^{z/l_s} + C_1 e^{Az}. \tag{B13}$$

At this point we assume that the upper boundary flux is purely advective. Physically this means we are imagining that the rate of surface erosion $E$, being externally imposed, removes $^{10}$Be atoms at a rate $En(h) = Wn(h)$. This is consistent with the requirement that the rate of quartz particle removal at the surface is equal to the rate at which quartz particles enter the soil at the soil-saprolite interface, independent of their $^{10}$Be concentration. Then, from Eq. (B8), $q_z(h) = P_0 l_s = Wn(h)$ so that

$n(h) = P_0 l_s / W$. Using this boundary condition,

$$C_1 = \frac{P_0 l_s}{W} e^{-Ah} - \frac{B l_s}{A l_s - 1} e^{h/l_s} e^{-Ah}. \tag{B14}$$

Substituting this into Eq. (B13) then doing algebra yields

$$n(z) = \frac{P_0 l_s}{W} e^{-W(h-z)/K_z} \tag{B15}$$

$$+ \frac{P_0 l_s^2}{K_z} \frac{1}{W l_s / K_z - 1} \left[ e^{-(h-z)/l_s} - e^{-W(h-z)/K_z} \right]. \tag{B15}$$

With dimensionless height $\hat{z} = z/h$, dimensionless concentration $\hat{n}(\hat{z}) = n(z)/n(h)$, primary Péclet number $Pe = Wh/K_z$ and secondary Péclet number $Pe_{l_s} = W l_s / K_z$, Eq. (B15) becomes

$$\hat{n}(\hat{z}) = e^{-Pe(1-\hat{z})}$$

$$+ \frac{Pe_{l_s}}{Pe_{l_s} - 1} \left[ e^{-h(1-\hat{z})/l_s} - e^{-Pe(1-\hat{z})} \right], \tag{B16}$$

which is Eq. (36) in the text.

## B2   Nonuniform mixing

With nonuniform mixing ($\kappa_z(z) = K_z z/h$) we start with Eq. (B9) with $C_1 = 0$ together with $q_z(z) = Wn(z) - K_z(z/h)\mathrm{d}n(z)/\mathrm{d}z$ to give

$$\frac{\mathrm{d}n(z)}{\mathrm{d}z} - \frac{Wh}{K_z z} n(z) = -\frac{P_0 l_s h}{K_z} e^{-h/l_s} \frac{e^{z/l_s}}{z}. \tag{B17}$$

We now simplify the notation and rewrite Eq. (B17) as

$$\frac{\mathrm{d}n(z)}{\mathrm{d}z} - \frac{A}{z} n(z) = -B \frac{e^{z/l_s}}{z}, \tag{B18}$$

with

$$A = \frac{Wh}{K_z} \quad \text{and} \quad B = \frac{P_0 l_s h}{K_z} e^{-h/l_s}. \tag{B19}$$

A general solution of Eq. (B18) is

$$n(z) = B\left(-\frac{z}{l_s}\right)^A \Gamma\left(-A, -\frac{z}{l_s}\right) + C_1 z^A. \tag{B20}$$

where $\Gamma$ is the incomplete gamma function. Using the advective boundary condition $n(h) = P_0 l_s / W$,

$$C_1 = \frac{P_0 l_s}{W} h^{-A} - B\left(-\frac{h}{l_s}\right)^A \Gamma\left(-A, -\frac{h}{l_s}\right) h^{-A}. \tag{B21}$$

Substituting this into Eq. (B20) then doing algebra yields

$$n(z) = \frac{P_0 l_s}{W}\left(\frac{z}{h}\right)^{Wh/K_z}$$

$$+\frac{P_0 l_s h}{K_z}\left[\left(-\frac{z}{l_s}\right)^{Wh/K_z}\Gamma\left(-\frac{Wh}{K_z}, -\frac{z}{l_s}\right)\right.$$

$$-\left(-\frac{h}{l_s}\right)^{Wh/K_z}$$

$$\left.\cdot\Gamma\left(-\frac{Wh}{K_z}, -\frac{h}{l_s}\right)\left(\frac{z}{h}\right)^{Wh/K_z}\right]e^{-h/l_s}. \tag{B22}$$

With dimensionless height $\hat{z} = z/h$, dimensionless concentration $\hat{n}(\hat{z}) = n(z)/n(h)$ and Péclet number $Pe = Wh/K_z$, Eq. (B22) becomes

$$\hat{n}(\hat{z}) = \hat{z}^{Pe} + Pe\left[\left(-\frac{h\hat{z}}{l_s}\right)^{Pe}\Gamma\left(-Pe, -\frac{h\hat{z}}{l_s}\right)\right.$$

$$\left(-\frac{h}{l_s}\right)^{Pe}\Gamma\left(-Pe, -\frac{h}{l_s}\right)\hat{z}^{Pe}\right]e^{-h/l_s}, \tag{B23}$$

which is Eq. (37) in the text. Note that Eq. (B23) has real and imaginary parts. Only the real part is physically meaningful in this problem.

Like the results in Section A1 above, the concentration gradient at the soil surface, $[\mathrm{d}n(z)/\mathrm{d}z]_h = 0$, as a consequence of assuming an advective boundary condition.

## Appendix C: Conservation of particles with finite OSL age

The number concentration $c(z,t)$ of particles with finite OSL age satisfies a Fokker-Planck-like equation, namely,

$$\frac{\partial c(z,t)}{\partial t} = -\frac{\partial}{\partial z}\left[w(z,t)c(z,t) - \kappa_z(z,t)\frac{\partial c(z,t)}{\partial z}\right], \tag{C1}$$

where $w(z,t)$ denotes the ensemble averaged particle velocity and $\kappa_z(z,t)$ denotes the ensemble expected particle diffusivity. For steady conditions involving a one-dimensional mean motion, $\partial c(z,t)/\partial t = 0$, $c(z,t) \to c(z)$, $w(z,t) \to W$, and Eq. (C1) becomes

$$\frac{\mathrm{d}}{\mathrm{d}z}\left[Wc(z) - \kappa_z(z)\frac{\mathrm{d}c(z)}{\mathrm{d}z}\right] = 0. \tag{C2}$$

Moreover, the particle flux must be zero across any surface normal to $z$, so

$$Wc(z) - \kappa_z(z)\frac{\mathrm{d}c(z)}{\mathrm{d}z} = 0. \tag{C3}$$

Under steady conditions the total number of particles with finite (measurable, non-saturated) OSL age within the soil element remains fixed. A particle entering the soil cannot attain a finite OSL age until it reaches the surface and is bleached, and then becomes buried and exposed to the dose field. Thus, even though particles that eventually possess a finite OSL age continuously

enter the element through its lower boundary, this boundary must be considered a zero flux boundary, as no particle with finite age can be added to the soil. Particles at the soil surface with zero OSL age are removed by erosion. The erosion rate matches $W$, so the rate of loss of particles is exactly balanced by the rate at which particles reach the surface and become OSL particles (with an OSL age of zero), many of which then take random walks downward. Thus, the upper boundary also must be considered a zero flux boundary with fixed concentration $c(h)$.

With uniform mixing ($\kappa_z = K_z$) we integrate Eq. (C3) to obtain

$$c(z) = C_1 e^{Wz/K_z}, \tag{C4}$$

with constant of integration $C_1$. The boundary condition $c(h) = C_1 e^{Wh/K_z}$ then leads to the solution

$$c(z) = c(h)e^{-W(h-z)/K_z}. \tag{C5}$$

The boundary condition $c(h)$ should be equal to the concentration of OSL sensitive particles entering the base of the soil

element, but which only take on finite OSL ages once they reach the surface and are bleached. With nonuniform mixing ($\kappa_z(z) = K_z z/h$) we integrate Eq. (C3) to obtain

$$c(z) = c(h)\left(\frac{z}{h}\right)^{Wh/K_z} \tag{C6}$$

With dimensionless height $\hat{z} = z/h$, dimensionless concentration $\hat{c}(\hat{z}) = c(z)/c(h)$ and Péclet number $Pe = Wh/K_z$, Eq. (C5) and Eq. (C6) become

$$\hat{c}(\hat{z}) = e^{-Pe(1-\hat{z})} \tag{C7}$$

and

$$\hat{c}(\hat{z}) = \hat{z}^{Pe}, \tag{C8}$$

which are Eq. (41) and Eq. (42) in the text. These results (Figure C1) are used next in obtaining the expected OSL age $A(z)$ of particles.

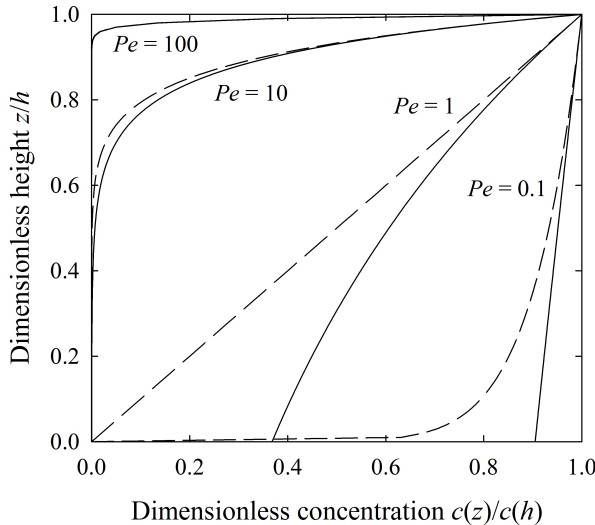

**Figure C1.** Plot of dimensionless OSL particle concentration $\hat{c}(\hat{z}) = c(z)/c(h)$ versus dimensionless height $\hat{z} = z/h$ with uniform mixing (solid lines) and nonuniform mixing (dashed lines) as these vary with the Péclet number $Pe = Wh/K_z$.

### Appendix D: Conservation of expected particle OSL age

Let $m_1(z,t)$ denote the expected (average) finite OSL age of particles within the small interval $z$ to $z + \mathrm{d}z$ in a soil element with dimensions $XYh$. With a total of $N$ such particles within the element, the product $Nc(z,t)m_1(z,t)XY\mathrm{d}z$ represents the total (collective) OSL age of particles within $\mathrm{d}z$. The product $c(z,t)m_1(z,t)$ satisfies a Fokker-Planck-like equation, namely,

$$\frac{\partial}{\partial t}[c(z,t)m_1(z,t)]$$

$$= -\frac{\partial}{\partial z}\bigg(w(z,t)c(z,t)m_1(z,t)$$

$$-\kappa_z(z,t)\frac{\partial}{\partial z}[c(z,t)m_1(z,t)]\bigg) + Sc(z,t), \tag{D1}$$

10    where $w(z,t)$ denotes the ensemble averaged particle velocity, $\kappa_z(z,t)$ denotes the ensemble expected particle diffusivity, and $S$ is a source term. For steady conditions in both $c$ and $m_1$ involving a one-dimensional mean motion, $(\partial/\partial t)[c(z,t)m_1(z,t)] = 0$, $c(z,t) \to c(z)$, $m_1(z,t) \to m_1(z)$, $w(z,t) \to W$, and Eq. (D1) becomes

$$\frac{\mathrm{d}}{\mathrm{d}z}\bigg(Wc(z)m_1(z) - \kappa_z(z)\frac{\mathrm{d}}{\mathrm{d}z}[c(z)m_1(z)]\bigg)$$

15    $$= Sc(z), \tag{D2}$$

We now rewrite Eq. (D2) as

$$\frac{\mathrm{d}}{\mathrm{d}z}\left(m_1(z)\left[Wc(z)-\kappa_z\frac{\mathrm{d}c(z)}{\mathrm{d}z}\right]\right.$$

$$\left.-\kappa_z(z)c(z)\frac{\mathrm{d}m_1(z)}{\mathrm{d}z}\right)=Sc(z)\,. \tag{D3}$$

Using (B3), this reduces to

$$\frac{\mathrm{d}}{\mathrm{d}z}\left[-\kappa_z(z)c(z)\frac{\mathrm{d}m_1(z)}{\mathrm{d}z}\right]=Sc(z)\,. \tag{D4}$$

indicating that the flux $q_z(z)=-\kappa_z(z)c(z)\mathrm{d}m_1(z)/\mathrm{d}z$ of particle OSL age involves only diffusion. That is, OSL age is not advected. In this problem, particles are advected, carrying their OSL age with them; but particle advection is balanced by particle diffusion.

We now write Eq. (D4) as

$$\frac{\mathrm{d}q_z(z)}{\mathrm{d}z}=Sc(z)\,. \tag{D5}$$

Vertically integrating,

$$\int_0^h\frac{\mathrm{d}q_z(z)}{\mathrm{d}z}\,\mathrm{d}z=S\int_0^h c(z)\,\mathrm{d}z\,. \tag{D6}$$

With fixed boundaries, this yields

$$q_z(h)-q_z(0)=S\bar{c}h\,. \tag{D7}$$

where the overbar denotes a vertically averaged quantity. Moreover, note that $q_z(0)=0$, as particles with a finite OSL age cannot be imported to the soil element. Thus,

$$q_z(h)=-\kappa_z(h)c(h)\left[\frac{\mathrm{d}m_1(z)}{\mathrm{d}z}\right]_{z=h}=S\bar{c}h\,. \tag{D8}$$

This indicates that OSL age is diffused "through" the soil surface at a rate equal to its total production within the soil column.

## D1 Uniform mixing

With uniform mixing ($\kappa_z=K_z$) we write

$$q_z(h)=S\int_0^h c(z)\,\mathrm{d}z\,. \tag{D9}$$

Using Eq. (C5), this becomes

$$q_z(h)=Sc(h)e^{-Wh/K_z}\int_0^h e^{Wz/K_z}\,\mathrm{d}z\,. \tag{D10}$$

Evaluating the integral then gives

$$q_z(h) = \frac{Sc(h)K_z}{W}\left(1 - e^{-Wh/K_z}\right). \tag{D11}$$

More generally,

$$q_z(z) = Sc(h)e^{-Wh/K_z}\int e^{Wz/K_z}\,\mathrm{d}z. \tag{D12}$$

5 Evaluating the integral,

$$q_z(z) = \frac{Sc(h)K_z}{W}e^{-Wh/K_z}e^{Wz/K_z} + C_1. \tag{D13}$$

Using the boundary condition obtained above for $q_z(h)$,

$$C_1 = -\frac{Sc(h)K_z}{W}e^{-Wh/K_z}. \tag{D14}$$

Using this result and $q_z(z) = -K_z c(z)\mathrm{d}m_1(z)/\mathrm{d}z$ with $c(z)$ given by Eq. (C5), we obtain

$$\frac{\mathrm{d}m_1(z)}{\mathrm{d}z} = -\frac{S}{W} + \frac{S}{W}e^{-Wz/K_z}. \tag{D15}$$

Integrating and using the boundary condition that $m_1(h) = 0$ then yields

$$m_1(z) = \frac{S}{W}(h - z)$$

$$+\frac{SK_z}{W^2}\left(e^{-Wh/K_z} - e^{-Wz/K_z}\right). \tag{D16}$$

15 With dimensionless height $\hat{z} = z/h$, dimensionless OSL age $\hat{m}_1(\hat{z}) = (W/h)m_1(z)$ and Péclet number $Pe = Wh/K_z$, Eq. (D16) becomes

$$\hat{m}_1(\hat{z}) = S(1 - \hat{z}) + \frac{Se^{-Pe}}{Pe}\left[1 - e^{Pe(1-\hat{z})}\right], \tag{D17}$$

which is Eq. (43) in the text.

### D2 Nonuniform mixing

20 With nonuniform mixing ($\kappa_z(z) = K_z z/h$) we use Eq. (C6) and write

$$q_z(h) = Sc(h)h^{-Pe}\int_0^h z^{Pe}\,\mathrm{d}z. \tag{D18}$$

Evaluating the integral then gives

$$q_z(h) = \frac{Sc(h)h}{1 + Pe}. \tag{D19}$$

More generally,

$$q_z(z) = Sc(h)h^{-Pe} \int z^{Pe}\,\mathrm{d}z\,. \tag{D20}$$

Evaluating the integral,

$$q_z(z) = \frac{Sc(h)h^{-Pe}}{1+Pe}z^{1+Pe} + C_1\,. \tag{D21}$$

Using Eq. (D19), $C_1 = 0$. Then using $q_z(z) = -K_z(z/h)c(z)\mathrm{d}m_1(z)/\mathrm{d}z$ together with Eq. (D19) we obtain

$$\frac{\mathrm{d}m_1(z)}{\mathrm{d}z} = -\frac{Sh}{K_z(1+Pe)}\,. \tag{D22}$$

Integrating and using the boundary condition that $m_1(h) = 0$ then yields

$$m_1(z) = \frac{Sh^2}{K_z(1+Pe)}\left(1 - \frac{z}{h}\right)\,. \tag{D23}$$

With dimensionless height $\hat{z} = z/h$, dimensionless OSL age $\hat{m}_1(\hat{z}) = (W/h)m_1(z)$ and Péclet number $Pe = Wh/K_z$, Eq. (D23) becomes

$$\hat{m}_1(\hat{z}) = \frac{SPe}{1+Pe}(1-\hat{z})\,, \tag{D24}$$

which is Eq. (44) in the text.

## Appendix E:  Variance of OSL ages

For a set of particles possessing finite OSL ages within any interval $\mathrm{d}z$, their rate of "aging" is fixed, independent of age. This means that the average OSL age increases at this fixed rate, whereas the second and higher moments do not change. Thus, direct production of the variance $m_2$ of OSL ages is zero.

For steady conditions we start with Eq. (40) in the text, namely,

$$\frac{\mathrm{d}}{\mathrm{d}z}\left[-\kappa_z(z)c(z)\frac{\mathrm{d}m_2(z)}{\mathrm{d}z}\right]$$

$$-2\kappa_z(z)c(z)\left[\frac{\mathrm{d}m_1(z)}{\mathrm{d}z}\right]^2 = 0\,. \tag{E1}$$

In the following we go directly to nondimensional forms of this.

## E1   Uniform mixing

With $\kappa_z = K_z$ and $\hat{q} = [h/K_z c(h)(h/W)^2]q$ we start with

$$\frac{\mathrm{d}\hat{q}}{\mathrm{d}\hat{z}} = \frac{\mathrm{d}}{\mathrm{d}\hat{z}}\left(-\hat{c}\frac{\mathrm{d}\hat{m}_2}{\mathrm{d}\hat{z}}\right) = 2\hat{c}\left(\frac{\mathrm{d}\hat{m}_1}{\mathrm{d}\hat{z}}\right)^2\,. \tag{E2}$$

Taking the derivative of Eq. (D17), squaring the result and using $\hat{c} = e^{-Pe}e^{Pe\hat{z}}$ then leads to

$$\frac{\mathrm{d}\hat{q}}{\mathrm{d}\hat{z}} = 2S^2 e^{-Pe}e^{Pe\hat{z}} - 4S^2 e^{-Pe} + 2S^2 e^{-Pe}e^{-Pe\hat{z}}. \tag{E3}$$

Integrating this with respect to $\hat{z}$ from $\hat{z} = 0$ to $\hat{z} = 1$ and noting that $\hat{q}(0) = 0$,

$$\hat{q}(1) = \frac{2S^2}{Pe} - 4S^2 e^{-Pe} - \frac{2S^2 e^{-2Pe}}{Pe}. \tag{E4}$$

More generally,

$$\hat{q}(\hat{z}) = \frac{2S^2 e^{-Pe}}{Pe}e^{Pe\hat{z}} - 4S^2 e^{-Pe}\hat{z}$$

$$-\frac{2S^2 e^{-Pe}}{Pe}e^{-Pe\hat{z}} + C_1. \tag{E5}$$

Using Eq. (E5) gives $C_1 = 0$. With $\hat{q} = -\hat{c}\,\mathrm{d}\hat{m}_2/\mathrm{d}\hat{z}$, and again using $\hat{c} = e^{-Pe}e^{Pe\hat{z}}$,

$$\frac{\mathrm{d}\hat{m}_2}{\mathrm{d}\hat{z}} = 4S^2 \hat{z}e^{-Pe\hat{z}} - \frac{2S^2}{Pe} + \frac{2S^2}{Pe}e^{-2Pe\hat{z}}. \tag{E6}$$

Integrating with respect to $\hat{z}$ and evaluatinng the constant of integration with the condition that $\hat{m}_2(1) = 0$ then yields

$$\hat{m}_2(\hat{z}) = \frac{2S^2}{Pe}(1 - \hat{z}) + \frac{4S^2}{Pe^2}\left[(1 + Pe)e^{-Pe}\right.$$

$$\left. - (1 + Pe\hat{z})e^{-Pe\hat{z}}\right] + \frac{S^2}{Pe^2}\left(e^{-2Pe} - e^{-2Pe\hat{z}}\right), \tag{E7}$$

which is Eq. (45) in the text.

## E2  Nonuniform mixing

With $\hat{q} = [h/K_z c(h)(h/W)^2]q$ we start with

$$\frac{\mathrm{d}\hat{q}}{\mathrm{d}\hat{z}} = \frac{\mathrm{d}}{\mathrm{d}\hat{z}}\left(-\hat{z}\hat{c}\frac{\mathrm{d}\hat{m}_2}{\mathrm{d}\hat{z}}\right) = 2\hat{z}\hat{c}\left(\frac{\mathrm{d}\hat{m}_1}{\mathrm{d}\hat{z}}\right)^2. \tag{E8}$$

With $\hat{c} = \hat{z}^{Pe}$ then $\hat{z}\hat{c} = \hat{z}^{1+Pe}$. Using this and taking the derivative of Eq. (D24) with respect to $\hat{z}$ and squaring the result leads

to

$$\frac{\mathrm{d}\hat{q}}{\mathrm{d}\hat{z}} = \frac{2S^2 Pe^2}{(1 + Pe)^2}\hat{z}^{1+Pe}. \tag{E9}$$

Integrating this with respect to $\hat{z}$ from $\hat{z} = 0$ to $\hat{z} = 1$ and noting that $\hat{q}(0) = 0$,

$$\hat{q}(1) = \frac{2S^2 Pe^2}{(2 + Pe)(1 + Pe)^2}. \tag{E10}$$

More generally,

$$\hat{q}(z) = \frac{2S^2 Pe^2}{(2 + Pe)(1 + Pe)^2} \hat{z}^{2+Pe} + C_1. \tag{E11}$$

Using Eq. (E10) gives $C_1 = 0$. With $\hat{q} = -\hat{z}\hat{c}\mathrm{d}\hat{m}_2/\mathrm{d}\hat{z}$, and again using $\hat{z}\hat{c} = \hat{z}^{1+Pe}$,

$$\frac{\mathrm{d}\hat{m}_2}{\mathrm{d}\hat{z}} = -\frac{2S^2 Pe^2}{(2 + Pe)(1 + Pe)^2} \hat{z}. \tag{E12}$$

5   Integrating with respect to $\hat{z}$ and evaluating the constant of integration with the condition that $\hat{m}_2(1) = 0$ then yields

$$\hat{m}_2(\hat{z}) = \frac{S^2 Pe^2}{(2 + Pe)(1 + Pe)^2}(1 - \hat{z}^2), \tag{E13}$$

which is Eq. (46) in the text.

*Author contributions.* All authors contributed to conceptualizing the problem and its technical elements. DF and RS contributed to the analytical analysis and the numerical simulations. RS wrote the final code. DF wrote much of the paper with contributions by RS and AK-Z.

10  *Competing interests.* We have no competing interests.

*Acknowledgements.* We acknowledge support by the U.S. National Science Foundation (EAR-1420831 to DJF and EAR-1734299 to RS). We appreciate continuing discussions with Peter Haff, Joshua Roering and Mark Schmeeckle concerning rarefied particle behavior, and Dan Morgan concerning cosmogenic radionuclide systematics.

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
