# Peer review of "The rarefied (non-continuum) conditions of tracer particle transport in soils, with implications for assessing the intensity and depth dependence of mixing from geochronology"

_Earth Surface Dynamics, 2018_

## Referee Comment (RC1) · Anonymous Referee #1 · 26 Sep 2018

This paper presents a detailed and rigorous treatment of when, why, and how to use probabilistic models to describe the transport of particulates in the landscape. In terms of understanding how soil particles migrate in natural settings, this work exposes and explores a number of ideas and concepts that benefit both theory and field practice.

Going forward I have a number of questions and suggestions that the authors may like to consider in making revisions to their work.

1. The authors present a full derivation of the governing transient Fokker-Planck equa-

tions. From a completeness point of view this derivation is appreciated but I feel that some of the main points the authors would like to make (e.g., the theoretical and practical advantages of probabilistic transport treatment over continuum treatments) get lost in the detail. I note that the actual theoretical, modeling, and practical calculations in the paper are all focussed on steady state problems, thus I would suggest that, while we may loose the full theoretical framework, just focusing on steady state problem might streamline the theory in the paper and shed a more focussed light on the take-home points the authors would like to make. One possible way to do this might be to put the rigorous developments of the equations into appendices and adapt the material currently in appendices A,B and C for the main text.

2. The authors do an excelled job of demonstrating the validity and utility of their probabilistic approach. The comparison and contrast of the theoretical (analytical) and numerical (random walk) solution for the mixing in a soil column provide convincing arguments that, for this class of problems, the probabilistic approach and its associated numerical model is an appropriate and flexible research tool. In my mind, there is little doubt that the proposed method is an excellent choice for analysis of mixing in soil columns.

Throughout the authors correctly note that their models are not continuum models and suggest critical differences between their probabilistic approach and treatments based on conventional (continuum) advection-diffusion equations. Never the less the governing equations used in the analysis in the paper, conservation of the expected number of 10Be atoms, conservation of particles with finite OSL age, and conservation of expected OSL age, are in "standard advection-diffusion forms". So essentially the difference between the probabilistic treatment and a continuum treatment, reduces in the construction of the velocity and diffusion terms and the definition of the dependent variables. Thus my question is: In a continuum treatment, that takes the soil particle properties and associates them with bulk soil properties, what are the possible forms of the dependent variables? Perhaps, explicitly identifying differences between probabilistic and continuum dependent variables, will provide additional illustration of the advantage of the probabilistic model?

3. In a more general context of land-scape process one might argue that a probabilistic transport model is always valid; if properly posed, such an approach will always reduce to a continuum model where appropriate. Never the less, in the spirit of reduced complexity modeling, can the authors comment on or provide guide-lines as to when a continuum approach would be reasonable. There is a hint of this on page 26 but could the authors expand and generalize?

4. There are a number of places in the paper where important concepts are presented. In some cases I feel that a figure may help to better illustrate the key ideas. In particular it may help to use a figure to illustrate the distinction between rarefied and continuous particle conditions—such a figure could be used in expanding guidelines for when to use a probabilistic model, see point 3 above.

5. The current paper is quite long but if the authors can streamline as I have suggested above they may also be able to add a summary/conclusion section.

Smaller points

Abstract: Line 1 put comers around "due to disturbance driven particle motions" Line 2 is it " the Fokker-Planck equation" or "a Fokker-Planck" equation? Line 6 The sentence starting "The analysis," may read better is it were split into two sentences

Page 4: Line 14 "target grains represent a subset of the total population of quartz grain sizes", what is the range of the total population

Page 10: Line 10 This line is a little confusing, why is eq. (14) an "advection equation"

Page 25: Line 7 "the formulation reveals that the expected particle OSL age (and the variance) satisfy a diffusion-like equation", what is the evidence for this?

[Figure]

2018.

---

## Referee Comment (RC2) · Anonymous Referee #2 · 6 Oct 2018

In this manuscript, Furbish, Schumer and Keen-Zebert investigate theoretically and numerically the evolution of tracer particles in a soil mobilized by all sorts of perturbations. More specifically, they address the concentration of cosmogenic isotopes such as $^{10}$Be and of quartz grains sensitive to Optically Stimulated Luminescence (OSL). To do so they derive, in much details, the evolution equations for the probability distribution of theses particles, using the formalism of the Fokker-Planck equations. They also run numerical simulations which reproduce the erratic motions of these particles. Finally, the authors comment on how their results could help the interpretation of field

measurements.

The manuscript is carefully written, and the mathematical derivations seem correct. I also believe that the authors' endeavor is needed, as more and more field data are collected, and often interpreted in a much wanting theoretical framework—in the literature, the $^{10}$Be concentration is often reported directly in terms of erosion rate, as if the two quantities were unequivocally related. I am therefore much supportive of the publication of this necessary work in E-surf.

The manuscript, however, is an arduous read. This is partly due to the subject itself, of course, but also, to a large extent, to the way the authors chose to present their results. Apart from the minor points listed below, I would like to encourage the authors to edit their manuscript to clarify their views.

Most of the difficulties stem from three decisions by the authors, namely (i) to present the mathematical derivations in full, even when they are very similar to each other, (ii) to treat the problems of OSL and of cosmogenic isotopes in parallel and (iii) to present their derivations and results first, and comment about them in the last section only, when the reader is most likely to have mixed them all together already.

I believe the authors could make the reading easier by presenting first the derivation of the Fokker-Planck equation for a cosmogenic isotope, and sending some intermediary steps of the derivations to the appendix. Indeed, these derivations are cumbersome, but not really difficult, and they often land on unsurprising expressions (equation (13) is a typical example). In most cases, the initial step of the derivation, and the necessary hypotheses, should suffice in the main document, before showing the final form of the expression. Then the authors could introduce their analytical solutions, and comment on them, before introducing the numerical simulations, and then comment on these new results.

Once the case of cosmogenic isotopes is clear, that of OSL particles would be easy to follow, if the authors content themselves with pointing at where the two derivations

differ from each other, and follow the order proposed above.

Regarding the interpretation of the results, I was also surprised to find a mixture of crucial statements, such as the one regarding the age of a fluid parcel (page 25), with merely tautological ones: "*particle properties are not in themselves subject to advection and diffusion, but rather, are merely carried with the particles*" (page 31)—how could it be otherwise?

Regarding the former, I would suggest making them more explicit, by writing in full what previous theories say, and compare it to the authors' results, ideally with a dedicated figure. As for the latter, I suspect the authors make these obvious claims because previous theories were ad odds with them. If it is so, I would recommend mentionning these theories explicitly, and point at where they might be flawed.

Overall, I would recommend spending more time on the points where the authors' theory differ from previous ones, especially on the ones that are accessible to measurements, and drop all secondary points, or send them to the appendix. I believe this would result in a clearer paper, thus doing justice to the authors' remarkable work.

Minor points

- Please write OSL in full in the abstract.

- Page 5, the discussion about the Knudsen number and the mean free path of particles is, at best, confusing. The Knudsen number compares the size of the system of interest to the mean free path of the particles, because the latter is the distance over which their trajectory looses its self-correlation. The value of the Knudsen number therefore tells us whether which macroscopic equations we can use, such as the Navier-Stokes ones. The density of passive tracers has nothing to do with this. A tracer particle interacts with many other particles before it encounters another tracer particle. That we follow a small number of

particles (large "*geometrical mean free path*") says nothing about the validity of the macroscopic equations. It affects the statistics of measurements only, which I believe is the point of the manuscript.

- Page 5, please define $V_p$ carefully. It took me some time before realizing what it is.

- Page 10, I found that calling $P$ an "advection speed" is confusing, and unnecessary. This is also true page 13.

- Page 20, are you sure the Fokker-Planck equations cannot tell us about the "*variability in $^{10}Be$ concentration of individual particles*"?

- Page 24, the discussion starting line 10 is confusing. Do you mean Lagragian vs. Eulerian? This point might be related to the one above. Please clarify.

- Page 29, the discussion about the disturbances starting line 10 seems interesting, but I could not really understand it. Would a simple example help?

- Page 35, the statement starting line 10 seems tautological to me. Please see my remark above.

---

## Author Comment (AC1) · 27 Oct 2018

27 October 2018

**General response:** We appreciate the efforts of the two referees in reviewing our work. Our revisions and responses below are based on careful consideration of their comments, questions and recommendations, and include the addition of an appendix as well as clarifying material in the text. Please note, however, that we

have opted to not reorganize the material as suggested. Our experience indicates that with a manuscript of this nature — involving a significant amount of mathematics with complementary numerical analyses — referee perspectives are likely to vary widely regarding presentation of the material, including differing opinions on what should be emphasized versus what could be deemphasized. Indeed, the suggestions for reorganization provided by the two referees are entirely disparate. As described in our responses below, we like our organization. Its progression is purposeful and systematic, including our choices of what should be emphasized (i.e., included in the main text) versus what is appropriately placed in appendixes. Our revisions therefore are focused on improving the clarity of the technical aspects of the work.

Please note that our references to page numbers below pertain to the original manuscript, not the revised manuscript.

**Referee #1 (Anonymous)**

This paper presents a detailed and rigorous treatment of when, why, and how to use probabilistic models to describe the transport of particulates in the landscape. In terms of understanding how soil particles migrate in natural settings, this work exposes and explores a number of ideas and concepts that benefit both theory and field practice.

Going forward I have a number of questions and suggestions that the authors may like to consider in making revisions to their work.

1. The authors present a full derivation of the governing transient Fokker-Planck equations. From a completeness point of view this derivation is appreciated but I feel that some of the main points the authors would like to make (e.g., the theoretical and practical advantages of probabilistic transport treatment over continuum treatments)

get lost in the detail. I note that the actual theoretical, modeling, and practical calculations in the paper are all focussed on steady state problems, thus I would suggest that, while we may loose the full theoretical framework, just focusing on steady state problem might streamline the theory in the paper and shed a more focussed light on the takehome points the authors would like to make. One possible way to do this might be to put the rigorous developments of the equations into appendices and adapt the material currently in appendices A, B and C for the main text.

We appreciate the motivation for this recommendation to place the formulation involving unsteady conditions in an appendix in order to streamline the presentation, given that the practical parts of the presentation are focused on steady conditions. Nonetheless, we prefer to retain the parts given to unsteady conditions within the main text for three reasons. First, although we focus on steady conditions in the end, we believe that there is value in presenting the more general formulation in anticipation that it might be useful to others interested in applying the ideas to unsteady problems, as mentioned in the Discussion and Conclusions. Second, the parts given to unsteady conditions are essential for illustrating the rationale and mathematical basis of the source/sink terms (Sections 3.2.2, 3.3.2 and 3.3.3) and how these terms, which represent rates, naturally arise from the probabilistic formulation. (In this regard, please see our response below concerning material on Page 10, which illustrates this second point.) Third, whereas we agree that placing part of the formulation in an appendix might streamline the presentation, we prefer to risk offering a slower read in that we view the more general formulation and the practical example as being of equal significance.

2. The authors do an excelled job of demonstrating the validity and utility of their probabilistic approach. The comparison and contrast of the theoretical (analytical) and numerical (random walk) solution for the mixing in a soil column provide convincing arguments that, for this class of problems, the probabilistic approach and

its associated numerical model is an appropriate and flexible research tool. In my mind, there is little doubt that the proposed method is an excellent choice for analysis of mixing in soil columns.

Throughout the authors correctly note that their models are not continuum models and suggest critical differences between their probabilistic approach and treatments based on conventional (continuum) advection-diffusion equations. Never the less the governing equations used in the analysis in the paper, conservation of the expected number of 10Be atoms, conservation of particles with finite OSL age, and conservation of expected OSL age, are in "standard advection-diffusion forms". So essentially the difference between the probabilistic treatment and a continuum treatment, reduces in the construction of the velocity and diffusion terms and the definition of the dependent variables. Thus my question is: In a continuum treatment, that takes the soil particle properties and associates them with bulk soil properties, what are the possible forms of the dependent variables? Perhaps, explicitly identifying differences between probabilistic and continuum dependent variables, will provide additional illustration of the advantage of the probabilistic model?

In our response here, we are not entirely sure that we are correctly interpreting these comments and questions, specifically regarding the idea of "differences between probabilistic and continuum dependent variable." Nonetheless, we recognize that they point to the need for further clarity regarding the probabilistic formulation of conservation in relation to rarefied versus continuum conditions. To this end we have added an appendix that uses familiar examples to elaborate key ideas in the text. The gist is this: The probabilistic formulation does not assume either rarefied or continuum conditions. It is indifferent to these conditions, yet equally applicable to both. As described in the text, the Fokker-Planck equation is a special case of the Master equation, a general statement of conservation of probability that is independent of scale, and which is the basis of more familiar statements of conservation when probability is reinterpreted

in terms of, for example, mass, momentum or energy. If in an individual system (realization) the continuum hypothesis is satisfied (a condition that is independent of the probabilistic basis of the Master equation or the Fokker-Planck equation), then the probabilistic formulation based on ensemble expected conditions and its continuum counterpart are essentially one and the same. If, however, the continuum hypothesis is not satisfied, then one cannot defensibly start with a continuum equation, but instead must appeal to a probabilistic formulation of ensemble expected conditions (in order to justify the use of continuously differentiable equations), with the proviso that any prediction of the behavior of an individual (rarefied) system is probabilistic in nature. Here we reemphasize a point introduced early in the development (Section 3.1), namely, "...despite the fact that Eq. (1) has the continuous form of a continuum advection-diffusion equation, Eq. (1) does not necessarily imply a continuum behavior. Only if conditions satisfy the continuum hypothesis can Eq. (1) be reinterpreted as an ordinary advection-diffusion equation describing transport and mixing in an individual (continuum) realization. For rarefied conditions, however, Eq. (1) represents the ensemble expected behavior, not necessarily what happens in an individual realization."

3. In a more general context of land-scape process one might argue that a probabilistic transport model is always valid; if properly posed, such an approach will always reduce to a continuum model where appropriate. Never the less, in the spirit of reduced complexity modeling, can the authors comment on or provide guide-lines as to when a continuum approach would be reasonable. There is a hint of this on page 26 but could the authors expand and generalize?

Following our response above, yes, a probabilistic formulation generally is valid; and, yes, such a formulation becomes equivalent to a continuum model under the right conditions. But to reiterate, whether or not the continuum hypothesis is satisfied is independent of the probabilistic formulation (e.g., the Fokker-Planck equation). Our

discussion on Page 26 does not actually address the question of continuum versus rarefied conditions, but instead is getting at the idea of how random variable quantities must be treated based on lessons learned from the probabilistic formulation. For example, "[w]hen the quantity of interest can be expressed as a total value within an interval $dz$, as with the total number of $^{10}$Be atoms, then this quantity may be treated as an intensive property of the bulk soil." But this result does not necessarily imply a continuum behavior. Thus, this discussion addresses the question of how to think about extensive versus intensive quantities that "belong" to the particles rather than to the bulk soil. We hope that the addition of an appendix as described above will help with the question concerning rarefied versus continuum conditions. Please also see our third response below to comments of Referee #2 regarding extensive and intensive quantities.

4. There are a number of places in the paper where important concepts are presented. In some cases I feel that a figure may help to better illustrate the key ideas. In particular it may help to use a figure to illustrate the distinction between rarefied and continuous particle conditions; such a figure could be used in expanding guidelines for when to use a probabilistic model, see point 3 above.

Please see our responses above. Here, too, we hope that the added appendix (with figures) will help.

5. The current paper is quite long but if the authors can streamline as I have suggested above they may also be able to add a summary/conclusion section.

Please see our responses to recommendations above.

Smaller points

Abstract: Line 1 put comers around "due to disturbance driven particle motion" Line 2 is it "the Fokker-Planck equation" or "a Fokker-Planck" equation? Line 6 The sentence starting "The analysis," may read better is it were split into two sentences

We have modified the Abstract.

Page 4: Line 14 "target grains represent a subset of the total population of quartz grain sizes", what is the range of the total population

The range of sizes of a population of quartz grains would depend on the source material, so we cannot offer a specific answer to this.

Page 10: Line 10 This line is a little confusing, why is eq. (14) an "advection equation"

A key phrase in this sentence is "...with respect to the $n_p$ domain..." To see why Eq. (14) is an advection equation, consider what might be a more familiar example. Let $f(x,t)$ denote a scalar quantity (e.g., concentration, temperature, etc.) that varies continuously with position $x$ and time $t$ within a continuum material that is moving with uniform speed $U$ parallel to $x$. Neglecting diffusion, let us now write

$$\frac{\partial f(x,t)}{\partial t} = -U\frac{\partial f(x,t)}{\partial x},$$

(1)

which is an advection equation that describes the rate at which $f(x,t)$ changes with respect to time as viewed from an Eulerian perspective. Note that this is a statement of conservation of the quantity $f(x,t)$. Now consider Eq. (14) in the text, namely,

$$\frac{\partial f_{V_p,n_p,z}(V_p,n_p,z,t)}{\partial t} = -P(z,t)\frac{\partial f_{V_p,n_p,z}(V_p,n_p,z,t)}{\partial n_p}.$$

(2)

Note that this is identical in form to Eq. (1) above with $f_{V_p,n_p,z}(V_p, n_p, z, t) \rightarrow f(x,t)$, $n_p \rightarrow x$ and $P(z,t) \rightarrow U$. Thus, Eq. (2) also is an advection equation that describes the rate at which the probability density $f_{V_p,n_p,z}(V_p, n_p, z, t)$ changes with time as viewed in an Eulerian sense with respect to the $n_p$ domain (rather than a spatial coordinate as in Eq. (1) above). Note that this is a statement of conservation of probability. Interestingly, although we do not show it here, Eq. (2) above is obtained from a Taylor expansion of the Master equation, and in effect is a Fokker-Planck equation without the diffusive term. We also note that a similar interpretation applies to Eq. (23) on Page 13 concerning advection with respect to the $A_p$ domain. This, too, is a statement of conservation of probability.

We prefer to not elaborate these points in the text, although we have added phrases to indicate that Eq. (14) and Eq. (23) are statements of conservation.

Here we offer an explanation of our description of $P(z,t)$ in Eq. (14) and of $S$ in Eq. (23) as being advective speeds in response to a comment below by Referee #2. The quantity $U$ in Eq. (1) above is unambiguously interpreted as a material speed (or velocity). However, because the Fokker-Planck equation is a statement of conservation of probability, the homologous quantity (e.g., $P$ or $S$) that appears in the advective term of this equation typically is referred to as a "drift speed" in the statistical mechanics literature, without necessarily connoting a material speed. (Nonetheless, see our parenthetical statement regarding Eq. (1) on Page 6, where the ensemble averaged velocity $w_p(z,t)$ indeed represents a material speed.) This reference just means the speed with which a quantity is being advected over the domain of interest — hence our use of advective speed to highlight that the probability density is being advected.

Page 25: Line 7 "the formulation reveals that the expected particle OSL age (and the variance) satisfy a diffusion-like equation", what is the evidence for this?

We have added a phrase to clarify that this is in reference to the appropriate diffusion equations appearing earlier in the text.

**Referee #2 (Anonymous)**

In this manuscript, Furbish, Schumer and Keen-Zebert investigate theoretically and numerically the evolution of tracer particles in a soil mobilized by all sorts of perturbations. More specifically, they address the concentration of cosmogenic isotopes such as $^{10}$Be and of quartz grains sensitive to Optically Stimulated Luminescence (OSL). To do so they derive, in much details, the evolution equations for the probability distribution of theses particles, using the formalism of the Fokker-Planck equations. They also run numerical simulations which reproduce the erratic motions of these particles. Finally, the authors comment on how their results could help the interpretation of field measurements.

The manuscript is carefully written, and the mathematical derivations seem correct. I also believe that the authors' endeavor is needed, as more and more field data are collected, and often interpreted in a much wanting theoretical framework — in the literature, the $^{10}$Be concentration is often reported directly in terms of erosion rate, as if the two quantities were unequivocally related. I am therefore much supportive of the publication of this necessary work in E-surf.

The manuscript, however, is an arduous read. This is partly due to the subject itself, of course, but also, to a large extent, to the way the authors chose to present their results. Apart from the minor points listed below, I would like to encourage the authors to edit their manuscript to clarify their views.

Most of the difficulties stem from three decisions by the authors, namely (i) to present the mathematical derivations in full, even when they are very similar to each other, (ii) to treat the problems of OSL and of cosmogenic isotopes in parallel and (iii) to present their derivations and results first, and comment about them in the last section only, when the reader is most likely to have mixed them all together already.

I believe the authors could make the reading easier by presenting first the derivation of the Fokker-Planck equation for a cosmogenic isotope, and sending some intermediary steps of the derivations to the appendix. Indeed, these derivations are cumbersome, but not really difficult, and they often land on unsurprising expressions (equation (13) is a typical example). In most cases, the initial step of the derivation, and the necessary hypotheses, should suffice in the main document, before showing the final form of the expression. Then the authors could introduce their analytical solutions, and comment on them, before introducing the numerical simulations, and then comment on these new results.

With regard to the last two comments above... First, this assessment (items (i), (ii) and (iii) above) of the organization we chose is correct; and we have retained this organization. Specifically, we chose to comment on the analytical and numerical results after presenting them in graphical form together (as opposed to commenting on the analytical results, then presenting and commenting on the numerical analyses, then reminding the reader of comments regarding the analytical results for comparison with the numerical results). In this manner the complementary analytical and numerical results immediately reinforce each other. Moreover, we "warn" the reader twice in the text (Pages 17 and 19) that the results are to be presented and commented on together so that the organization does not come as a surprise. Turning to the issue of presenting material in the text versus appendixes, we first note that currently the appendixes in the one-column format of the paper are 10 pages (before addition of a new appendix that is about five pages). The "offending" text with derivations is

about four pages (or 11% of the main text). In our view, these derivations contain key elements and steps that illustrate the proper treatment of intensive and extensive quantities, which we prefer to keep in the main text rather than treating them as being of secondary value relative to the starting and ending points of the derivations. Indeed, it is important to us to demonstrate how and why a carefully posed probabilistic treatment of intensive and extensive quantities yields an "unsurprising" result, as this lends confidence that the formulation is sound. That Eq. (13) is perceived as being unsurprising suggests that we achieved our objective, because aside from its advection-diffusion form, we previously have seen nothing like it in the literature. (And please see our response below to the next related comment.) We recognize that this is a lengthy paper. But we prefer that it not become a lengthy series of appendixes. Moreover, virtually all intermediate derivations associated with the steady conditions described in Section 4.1 and 4.2 were originally placed in appendixes, as these mostly represent the steps involved in solving the differential equations rather than revealing any key elements of the formulation as described above.

Once the case of cosmogenic isotopes is clear, that of OSL particles would be easy to follow, if the authors content themselves with pointing at where the two derivations differ from each other, and follow the order proposed above.

This is in fact precisely what we did. To wit, on Page 12 in the original manuscript we offered the starting point for OSL ages (showing how it differs from the starting point for $^{10}$Be) and then stated... "With Eq. (20) and Eq. (21) in place, we multiply both by $A_p$, integrate with respect to $A_p$, *then follow the same steps as presented in Section 3.2.1 above* to give..." And on page 13, after offering the starting point, we stated... "We then multiple Eq. (23) by $A_p$, integrate with respect to $A_p$, *then follow the same steps as presented in Section 3.2.2 above* to give..." We suggest that these references to previously presented material obviated approximately four pages of derivations.

Regarding the interpretation of the results, I was also surprised to find a mixture of crucial statements, such as the one regarding the age of a fluid parcel (page 25), with merely tautological ones: "*particle properties are not in themselves subject to advection and diffusion, but rather, are merely carried with the particles*" (page 31) — how could it be otherwise?

This is not merely tautological. We are emphasizing the following points developed previously in the text. Namely, imagine "dissolving" $^{10}$Be atoms within a continuum fluid (e.g., water). These atoms are just a subset of all "particles" making up the continuum fluid and therefore "belong to" the fluid. Their concentration is defined by the number of atoms per unit continuum volume, and they undergo molecular-scale diffusion (and dispersion if flow is turbulent) as normally envisioned for dissolved materials within a continuum fluid. As well, they may be advected as normally envisioned. But now consider instead an individual quartz particle with a specific number of $^{10}$Be atoms within it, where this number of atoms varies from one particle to another. If these quartz particles are "suspended" within a continuum, then the $^{10}$Be atoms do not diffuse within the continuum as normally envisioned, as they belong to the particles, not the continuum. One must therefore treat the problem as we did, determining expected values of *particle* $^{10}$Be concentrations, then specify the variations in these expected values as the particles (with different $^{10}$Be concentrations) undergo advection and diffusion. This is in contrast to specifying at the outset the $^{10}$Be concentration as a number of atoms per unit continuum volume (as though they belong to the continuum). Similar comments apply to expected particle OSL ages, particularly because "...the expected number concentration... of particles possessing a finite OSL age generally is not uniform over $z$." We agree that it could not be otherwise. In addition, we note that on Page 31 we are referring specifically to advection over the $n_p$ and $A_p$ domains as a contrast to the idea that "...extensive and intensive particle properties are... carried with the particles as these undergo advection and diffusion with respect to $z$."

Regarding the former, I would suggest making them more explicit, by writing in full what previous theories say, and compare it to the authors' results, ideally with a dedicated figure. As for the latter, I suspect the authors make these obvious claims because previous theories were ad odds with them. If it is so, I would recommend mentionning these theories explicitly, and point at where they might be flawed.

We appreciate the sentiment of this suggestion. However, given the variety of descriptions of soil particle mixing (and the treatment of extensive and intensive particle properties) in the literature, here we prefer our approach of offering a thorough, defensible treatment of the problem, and moving forward. We purposefully avoided adding 'a critique of previous work' to the five stated objectives of our paper.

Overall, I would recommend spending more time on the points where the authors' theory differ from previous ones, especially on the ones that are accessible to measurements, and drop all secondary points, or send them to the appendix. I believe this would result in a clearer paper, thus doing justice to the authors' remarkable work.

Please see our preceding responses.

Minor points

• Please write OSL in full in the abstract.

We have modified the Abstract.

• Page 5, the discussion about the Knudsen number and the mean free path of particles is, at best, confusing. The Knudsen number compares the size of the system of interest to the mean free path of the particles, because the latter is the distance over

which their trajectory looses its self-correlation. The value of the Knudsen number therefore tells us whether which macroscopic equations we can use, such as the Navier-Stokes ones. The density of passive tracers has nothing to do with this. A tracer particle interacts with many other particles before it encounters another tracer particle. That we follow a small number of particles (large "*geometrical mean free path*") says nothing about the validity of the macroscopic equations. It affects the statistics of measurements only, which I believe is the point of the manuscript.

This interesting comment regarding the significance of the Knudsen number suggests a need for clarification. We start our response by noting that a Knudsen number that "compares the size of the system of interest to the mean free path" is one of several ways to define this number, and is particularly relevant when considering particle flows within small, confined spaces. Another useful definition of the Knudsen number involves the ratio of the mean free path to a computational grid size or measurement interval, and is relevant in calculations aimed at assessing effects of grid sizes or measurement intervals (see new Appendix A). Yet another involves the gradient length scale. The mean free path in the numerator of the Knudsen number as applied to gases simultaneously reflects two interrelated quantities: the frequency of particle collisions and the number density of the particles. (The mean free path can be written in terms of either of these.) The former is important for assessing whether a probabilistic kinetic description (e.g., the Boltzmann equation) justifiably gives way to a macroscopic description involving continuum quantities (i.e., state variables and related constitutive relations) at a specified length scale $L$. The latter is equally important for imagining whether macroscopic quantities vary smoothly when viewed at a continuum scale (i.e., upon satisfying the idea of a "representative elementary volume" or "physical point"). We agree that the geometrical mean free path cannot be interpreted in the same way as the mean free path as defined for a gas, notably because the kinetics are not relevant, and, as the referee points out, "a tracer particle interacts with many other particles before it encounters another tracer particle." But as

a measure of number density the mean free path has much to say about the scales at which the tracer particle concentration can be viewed as varying smoothly with position, and therefore whether the Fokker-Planck equation represents the behavior of an individual realization that satisfies the continuum hypothesis versus an expected behavior for rarefied conditions. So... we have slightly revised the text to point out that our description of the Knudsen number with respect to a gas is to emphasize this as an example of a continuum condition as conventionally defined, and that our transition to the mean spacing for tracer particles is homologous to the idea of the mean free path concerning rarefied versus continuum conditions, absent reference to kinetic behavior. We also believe that the added Appendix A will help.

• Page 5, please define $V_p$ carefully. It took me some time before realizing what it is.

We have added *individual* particle volumes $V_p$ to clarify.

• Page 10, I found that calling $P$ an "advection speed" is confusing, and unnecessary. This is also true page 13.

Please see our explanation above (in response to a comment by Referee #1) concerning the use of "speed" in this context. We have retained this wording, given the probabilistic context and the explanations offered in the sentences, but we removed the quotations. This is now consistent with our parenthetical statement regarding the interpretation of $w_p(z,t)$ appearing in Eq. (1) on Page 6.

• Page 20, are you sure the Fokker-Planck equations cannot tell us about the "*variability in* $^{10}Be$ *concentration of individual particles*"?

Interesting! Within the context of our statements surrounding this phrase, it is

entirely appropriate as written. That said, indeed we could have formulated a version of the Fokker-Planck equation to describe the second and higher moments of particle [10]Be concentrations, as we did for the second moment of particle OSL ages because of the significance of this moment in relation to sampling. So in this respect, yes, a Fokker-Planck equation could be used to reveal information about variability in [10]Be concentrations of individual particles. However, with reference to to Figure 5 for [10]Be (and to Figure 8 for OSL), low-order moments capture limited information regarding the forms of the distributions depicted in these figures, and of the associated effects of these forms in relation to advection over the $n_p$ and $A_p$ domains as described in the text. And, perhaps more importantly, knowledge of the behavior of the second and higher moments of [10]Be concentrations of individual particles as revealed by a Fokker-Planck formulation is not particularly useful, as individual particle concentrations are not measured. (To our knowledge, individual particle concentrations cannot be measured with current techniques.)

• Page 24, the discussion starting line 10 is confusing. Do you mean Lagragian vs. Eulerian? This point might be related to the one above. Please clarify.

The material in this paragraph is not referring to a Lagrangian versus an Eulerian perspective. This paragraph is a summary statement of ideas previously described in similar terms at several points in the text. We prefer to retain the current wording

• Page 29, the discussion about the disturbances starting line 10 seems interesting, but I could not really understand it. Would a simple example help?

We have added key wording to these two paragraphs to elaborate the conceptualization of the effects of disturbance scales and mixing.

• Page 35, the statement starting line 10 seems tautological to me. Please see my remark above.

Despite several careful re-reads, we see nothing at this location in the text that could be interpreted as a tautology.

**Addendum**

Please note that we have: 1) removed the word "vertical" from the sentence containing "...may vary with its [vertical] position, and therefore with time..." on Page 16, as any small motion of a particle can change its micro-dosimetry; and 2) added references to Parker and Perg (2005) and von Smoluchowski (1906).

\*\*\*\*\*\*\*\*\*

DJF, RS and AK-Z